# Binder driven self-assembly of metal-organic cubes towards functional hydrogels

Papri Sutar[1], Venkata M. Suresh[1], Kolleboyina Jayaramulu [1], Arpan Hazra[1] & Tapas Kumar Maji[1]

The process of assembling astutely designed, well-defined metal-organic cube (**MOC**) into hydrogel by using a suitable molecular binder is a promising method for preparing processable functional soft materials. Here, we demonstrate charge-assisted H-bonding driven hydrogel formation from $Ga^{3+}$-based anionic **MOC** $((Ga_8(ImDC)_{12})^{12-})$ and molecular binders, like, ammonium ion $(NH_4^+)$, N-(2-aminoethyl)-1,3-propanediamine, guanidine hydrochloride and $\beta$-alanine. The morphology of the resulting hydrogel depends upon the size, shape and geometry of the molecular binder. Hydrogel with $NH_4^+$ shows nanotubular morphology with negative surface charge and is used for gel-chromatographic separation of cationic species from anionic counterparts. Furthermore, a photo-responsive luminescent hydrogel is prepared using a cationic tetraphenylethene-based molecular binder (DATPE), which is employed as a light harvesting antenna for tuning emission colour including pure white light. This photo-responsive hydrogel is utilized for writing and preparing flexible light-emitting display.

[1] Molecular Materials Laboratory, Chemistry and Physics of Materials Unit, School of Advanced Materials (SAMat), Jawaharlal Nehru Centre for Advanced Scientific Research, Bangalore 560064, India. These authors contributed equally: Papri Sutar and Venkata M. Suresh. Correspondence and requests for materials should be addressed to T.K.M. (email: tmaji@jncasr.ac.in)

Charge-assisted hydrogen bond (CAHB) is a type of non-covalent interaction ($X-H^{(+)}...Y^{(-)}$) that plays an important role in the structure-property correlation of bio-macromolecules and in various biological molecular recognition processes[1–4]. CAHB is also widely employed in the construction of discrete organic cages, extended crystalline metal-organic architectures[5,6] and soft supramolecular gels[7]. The reason behind such versatility of CAHB is essentially its intrinsic strength (stronger than neutral X-H···Y bond) and directionality, that results in a wide range of materials with an array of exciting and complementary properties[8]. In this regard, CAHB driven self-assembly of predesigned metal-organic polyhedra (MOPs)[9–29] that are discrete metal-organic cages with confined cavities and large number of connecting sites, into soft supramolecular hydrogel is yet to accounted.

Among different classes of MOPs, metal organic cubes (MOCs) with a general formula $[M_8L_{12}]^x$ ($x = 0$, n-), comprising metal ions ($M^{n+} = Ni^{2+}$, $Zn^{2+}$, $In^{3+}$, $Cr^{3+}$) as vertices and imidazoledicarboxylate (L) as edges of a cube have been well explored[30]. Aesthetic appeal, structural modularity and robustness pertaining to the MOCs showed great promise for diverse applications[31]. MOCs are neutral ($x = 0$) or anionic ($x = n-$, a-MOC) depending upon the charge balance between $M^{n+}$ and $L^{32}$. MOCs are exploited as molecular building blocks by connecting the peripheral free carboxylate oxygens with metal ions or with H-bond donor molecules and the resulting extended structures showed potential applications in gas storage/ separation and proton-conductivity[33–35]. Since exteriors of a-MOCs are decorated with free polar carboxylate groups, they could be soluble in polar solvents, like water[36]. We envisioned that interaction of soluble a-MOCs with the positively charged or neutral, H-bond donor molecular binders through CAHB interaction could result in extended, supramolecular network[37,38]. In aqueous solution, such supramolecular assembly between a-MOC and different molecular binders could result in hydrogels. Aida et al. have shown that CAHB interaction between anionic clay nanosheets and dendritic molecular binders containing multiple guanidinium ions facilitated the cross-linking of the clay nanosheets and formed hydrogels[39]. Recently, Johnson et al. and Nitschke et al. reported the self-assembly of soluble polymers having coordinating end groups with metal ions, leading to the formation polymeric gel that consisted of in-situ formed metal-organic cage at the junction of cross-linked polymers[40–42].

Although their approach is inspiring, use of water soluble, preformed a-MOC as a platform to study self-assembly in the presence of different molecular binders is yet to be accounted. We envision that introduction of different molecular binders would tune the nano-morphologies and functionalities of the a-MOC-hydrogels. For example, the surface charge of the hydrogel-nanostructure could be altered by choosing appropriate cationic/ anionic binders, making the hydrogel useful for chromatographic separation of oppositely charged species. Moreover, suitably designed chromophoric molecular binder would result in a processable soft luminescent hybrid hydrogel. The a-MOCs could also act as an excellent template for immobilizing the multi-chromophoric donors and acceptor binders for light harvesting application. In such system the emission property can be tuned and even processable high quantum efficiency pure white-light-emitting materials can be realized. In addition, stimuli responsive molecular binders can also provide the stimuli-responsive a-MOC-hybrid gels. With the introduction of photoactive molecular binders, photo-responsive hydrogel could also be prepared, which can enable writing on the flexible displays by change in corresponding photochemical reactions.

Herein, we report synthesis of a $Ga^{3+}$ based metal-organic cube, $((Me_2NH_2)_{12}(Ga_8(ImDC)_{12})\cdot DMF\cdot 29H_2O)$ (1), extended into three dimension through CAHB interaction between anionic $[Ga_8(ImDC)_{12}]^{12-}$ (MOC) and $Me_2NH_2^+$ (DMA) cations. Compound 1 is highly soluble in water, where discrete MOCs remain intact in solution. This particular phenomenon provides an opportunity to crosslink the MOCs with a wide range of molecular binders that lead to the formation of charge-assisted hydrogels (Fig. 1a). Different molecular binders are assembled with MOC and the resulting hydrogels show different morphologies and properties (Fig. 1b). When ammonium cation ($NH_4^+$) is used as molecular binder, the resulting hydrogel with negatively charged, tubular nanostructures is exploited for gel chromatographic separation of positively charged species. We also extend the concept to form stimuli responsive luminescent hydrogel by rationally designing an aggregation induced emission (AIE)-active molecular binder containing tetraphenylethene (TPE) core. The photoresponsive behaviour of this hydrogel is further exploited for writing on flexible displays based on photocyclization of TPE core. Such photoresponsive behaviour of the hydrogel is also explored for tuning the excitation energy transfer from TPE segment to encapsulated acceptor dye. Finally, a pure white-light-emitting hydrogel with Commission Internationale de L'Eclairage (CIE) co-ordinates of (0.33, 0.32) is achieved.

## Results

**Synthesis and structural characterization of** 1. Solvothermal reaction of $Ga(NO_3)_3\cdot 6H_2O$ and 4,5-imidazoledicarboxylic acid ($H_3ImDC$) in $N,N'$-dimethylformamide (DMF) in presence of triethylamine ($Et_3N$) at 120 °C affords a pale yellow powder. Aqueous solution of the powder on slow evaporation yields block-shaped single crystals of 1. The asymmetric unit contains two $Ga^{3+}$ (Ga1, Ga2) centres, two $ImDC^{3-}$, two dimethyl ammonium cations ($Me_2NH_2^+$, DMA) and eight guest water and one DMF molecules (Supplementary Figure 4 and Supplementary Table 1). The DMA cations are formed in-situ from DMF under solvothermal condition[43,44]. Each $ImDC^{3-}$ (ImDC_a or ImDC_b) linker chelates two $Ga^{3+}$ (Ga1···Ga2 or Ga1···Ga1) centres in a bis(bidentate) fashion through two imidazole nitrogen atoms (N1, N2 or N3, N4) and two carboxylate oxygen atoms (O1, O2 or O5, O8), while other two oxygen atoms (O3, O4 or O6, O7) remain free (Fig. 2a and Supplementary Figure 4). Twelve $ImDC^{3-}$ alternatively connect eight $Ga^{3+}$ centres to form an anionic metal-organic cube, $(Ga_8(ImDC)_{12})^{12-}$ (MOC), where Ga1···Ga2 and Ga1···Ga1 distances are 6.232 Å and 6.244 Å, respectively, and ∠Ga1···Ga2···Ga1, ∠Ga1···Ga1···Ga2, ∠Ga1···Ga1···Ga1 angles are 90.11°, 90.00°, 89.89°, respectively (Supplementary Figure 5, Supplementary Table 2). In the $(Ga_8(ImDC)_{12})^{12-}$ cube eight $Ga^{3+}$ ions occupy the vertices and twelve $ImDC^{3-}$occupy the edges (Fig. 2a and Supplementary Figure 5). The negative charge from 12 carboxylate group of each cube is neutralized by twelve surrounding DMA cations. Two DMA cations (N5 and N7) play an important role in extending of MOCs as they connect two adjacent cubes through N-H···O H-bonding interaction with pendent carboxylate oxygen atoms (O3, O4 and O7, O6) (Fig. 2b and Supplementary Figure 5). The N-H···O H-bond distances and ∠N-H···O H-bond angles are in the range of 1.992–2.127 Å and 127–166°, respectively, which indicate the presence of strong charge-assisted intermolecular H-bonding interaction. The minimum distance between two adjacent cube (O3···O7) is 3.429 Å. Each cube is concomitantly connected to six neighbouring ones through H-bonding with DMA cations and forming a 3D extended structure (Fig. 2c). The periodic arrangement of cubes generates an open framework which exhibits two types of alternative 3D channels with an approximate window size of (5.9 × 4.4 Å$^2$) and (5.4 × 0.18 Å$^2$), respectively (Fig. 2d). These 3D channels are occupied by disordered

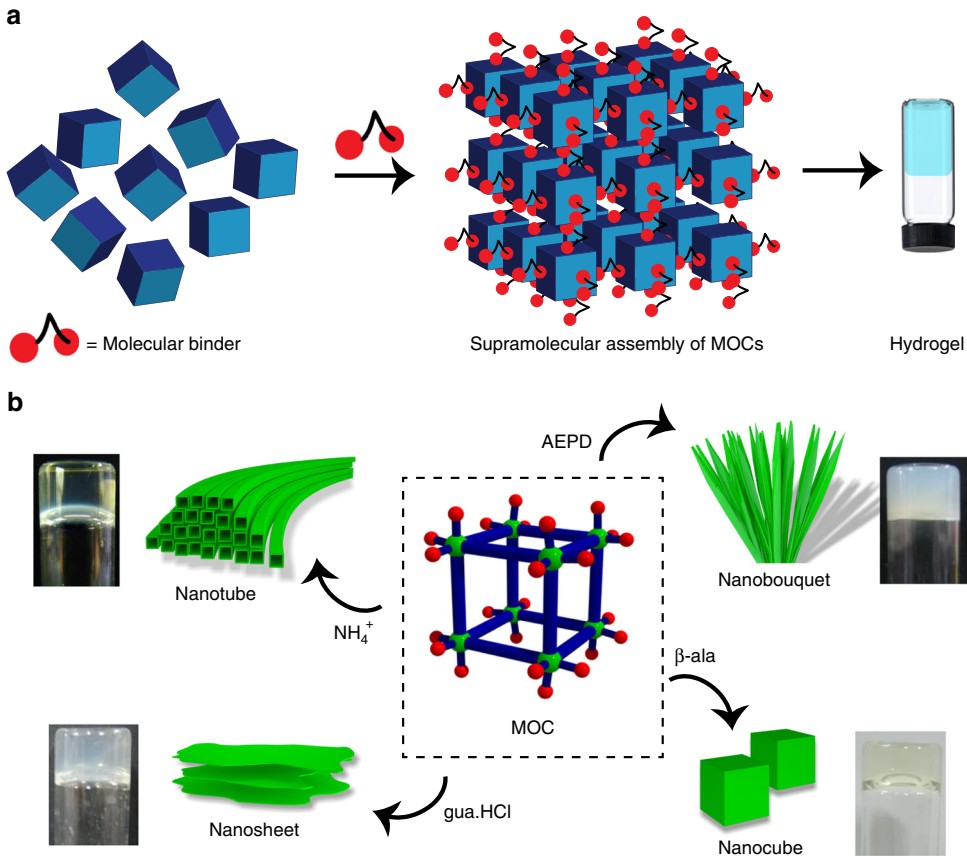

**Fig. 1** Molecular binder-driven self-assembly of **MOC** to hydrogels. **a** Schematic representation of CAHB driven self-assembly of **MOC** with small molecular binders towards the formation of hydrogel. **b** Self-assembly of **MOC** with different molecular binders ($NH_4^+$ cation, AEPD, gua.HCl and β-ala) results in hydrogels with nanotube, nanobouquet, nanosheet and nanocube morphologies, respectively

water and DMF guest molecules (Supplementary Figure 6). Thermogravimetric analysis (TGA) of as-synthesized **1** shows an initial weight-loss of 15% at 200 °C that corresponds to a loss of guest water and DMF molecules (Supplementary Figure 7). The similar powder X-ray diffraction (PXRD) pattern of the as-synthesized powder and the simulated one indicates purity of the compound (Supplementary Figure 8). $N_2$ adsorption isotherm of desolvated **1** (**1′**) at 77 K shows a type-II profile indicating surface adsorption (Supplementary Figure 9). However, $CO_2$ adsorption isotherm of **1′** at 195 K shows type I behaviour with the total uptake of ~75 mL g$^{-1}$ suggesting microporous nature of the extended framework of **1** (Supplementary Figure 9).

**Preparation and characterization of MOC-G1 hydrogel**. **1** is highly soluble in water (Supplementary Figure 10). The negative ion acquisition mode high resolution mass spectra (HRMS) of the aqueous solution shows three peaks at $m/z = 2467.32$ ($z = 1-$), 1233.16 ($z = 2-$) and 822.58 ($z = 3-$) corresponding to $((Ga_8(ImDC)_{12})(9\,H^+)(2Na^+)(H_2O))^-$, $((Ga_8(ImDC)_{12})(8\,H^+)(2Na^+)(H_2O))^{2-}$, and $((Ga_8(ImDC)_{12})(7\,H^+)(2Na^+)(H_2O))^{3-}$ moieties, respectively (Supplementary Figure 11). This confirms high stability of **MOC** in the aqueous solution. The presence of negatively charged carboxylate oxygen on the periphery of **MOC** ($z = 12-$) makes it polar and highly soluble in water. Moreover, $^1H$-NMR spectrum of **1** in $D_2O$ shows the presence DMA cations (Supplementary Figure 12). Addition of 3.5% aq. $NH_3$ into the aqueous solution of **1** results in stable, transparent hydrogel (**MOC-G1**) after 8 h at room temperature (critical gelation concentration = 15 mg ml$^{-1}$) (Fig. 2e). Notably, no gelation is

observed when aq. $NH_3$ is added into the aqueous solution of ligand (4,5-imidazoledicarboxylic acid), confirming importance of anionic **MOC** in hydrogel formation (Supplementary Figure 13). **MOC-G1** does not show any visible weakening over a month. The sol-gel transition is completely reversible after multiple shaking-resting cycles, indicating its thixotropic behaviour. Upon heating, the **MOC-G1** loses entrapped water molecules and does not show thermo-reversibility. However, **MOC-G1** exhibits excellent pH responsive behaviour. When 0.1 N HCl (pH = 4–5) is added to **MOC-G1** (intrinsic pH = 11) a precipitate forms which reforms gel after addition of aq. $NH_3$ (pH = 12) (Supplementary Figure 14). This indeed suggests the interaction of $NH_4^+$ with **MOC** is crucial for hydrogelation and it is feasibly driven through charge-assisted H-bonding between $NH_4^+$ and peripheral carboxylate oxygens of **MOC** (Fig. 2f). PXRD of **MOC-G1** xerogel shows similar Bragg's reflections as observed in **1**, indicating **MOCs** are intact in gel (Supplementary Figure 15). The stability of the **MOC** in **MOC-G1** hydrogel is further confirmed by negative ion acquisition mode HRMS analysis which shows peaks at $m/z = 2469.64$ ($z = 1-$), 1233.36 ($z = 2-$) and 822.23 ($z = 3-$) corresponds to $((Ga_8(ImDC)_{12})(9\,H^+)(2Na^+)(H_2O))^-$, $((Ga_8(ImDC)_{12})(8\,H^+)(2Na^+)(H_2O))^{2-}$ and $((Ga_8(ImDC)_{12})(7\,H^+)(2Na^+)(H_2O))^{3-}$ moieties, respectively (Supplementary Figure 16). Rheology studies of **MOC-G1** reveal that storage modulus ($G'$) is higher than the loss modulus ($G''$) and both of them ($G'$ and $G''$) are independent of angular frequency ($\omega$) over a large range of strain (%) indicating viscoelastic nature of the hydrogel (Fig. 2g and Supplementary Figure 17). The value of $G'$ at high strain (%) is 52.4 Pa. The stress versus strain plot shows the yield stress and yield strain values as 6.18 Pa and 17.8%,

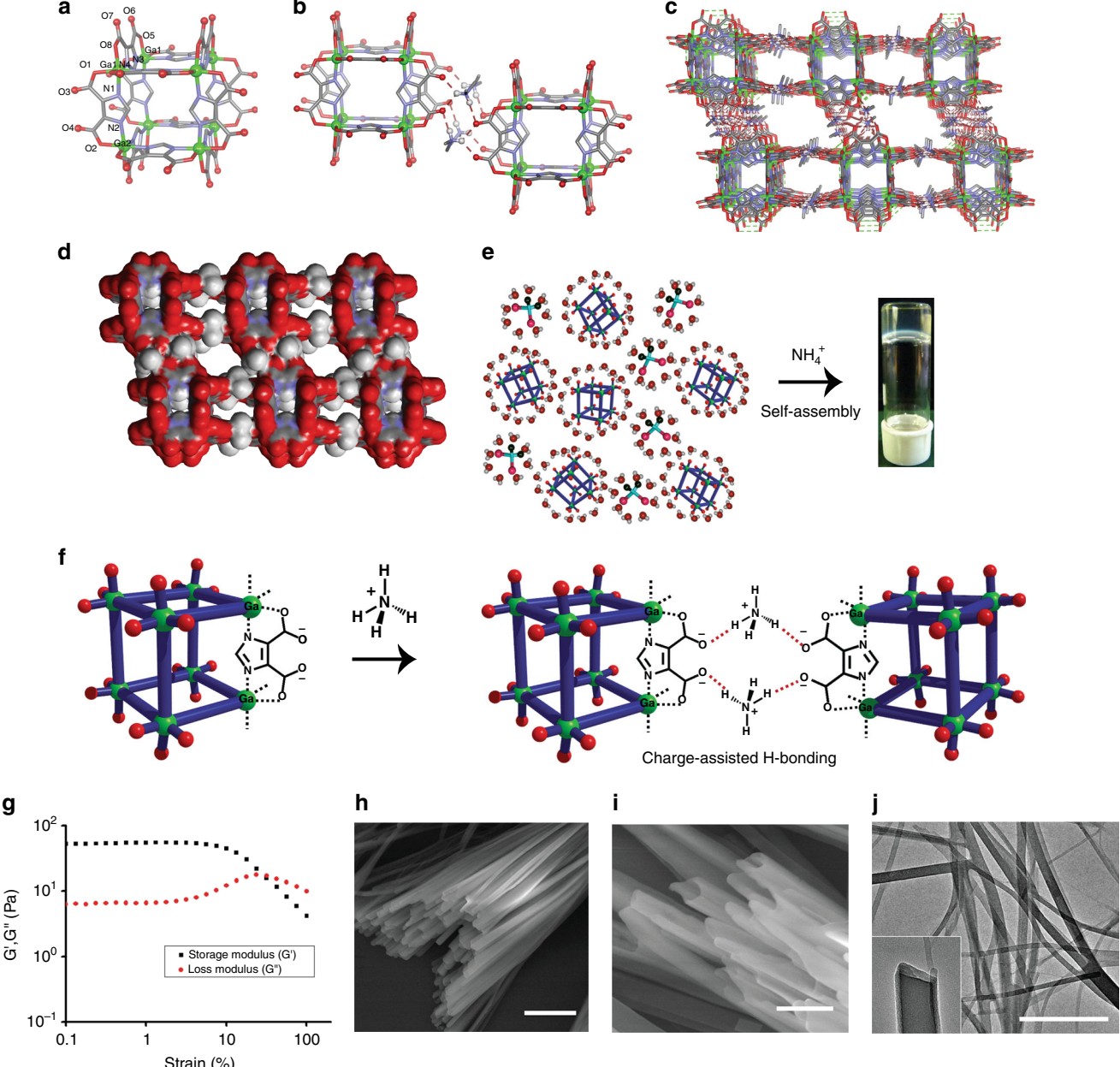

**Fig. 2** Crystal structure of **1** and self-assembly of **MOC** with $NH_4^+$ binder. **a** Structure of $(Ga_8(ImDC)_{12})^{12-}$ cube (**MOC**). **b** Two **MOC**s are connected to each other via intermolecular H-bonding with four DMA cations. **c** 3D supramolecular packing of the **MOC**s in **1**, where each cube is connected to six neighbouring cubes through DMA cations. **d** Space filling structure of **1** showing the window of the 3D channels. **e** Formation of **MOC-G1** hydrogel from aq. solution of **MOC**s in presence of $NH_4^+$ binder. **f** Schematic showing self-assembly of **MOC** with $NH_4^+$ binder through charge-assisted H-bonding interaction. **g** Oscillatory strain measurements (frequency = 1.0 rad/s) of **MOC-G1**, the squares (black) and circles (red) indicate storage ($G'$) and loss modulus ($G''$), respectively. **h, i** FESEM images of **MOC-G1** xerogel showing the formation of nanotubes with rectangular cross-section, **h** scale bar = 2 μm and **i** scale bar = 1 μm. **j** TEM image of **MOC-G1** xerogel showing the formation of nanotubes (scale bar = 500 nm). Inset is showing opening of the hollow nanotube

respectively (Supplementary Figure 17). Field emission scanning electron microscope (FESEM) images of **MOC-G1** xerogel show micrometer (3 μm–8 μm) long tubular nanostructures with rectangular cross-section (Fig. 2h, i and Supplementary Figure 18). The transmission electron microscopy (TEM) analysis indicates that the diameter of inner channel is 80–100 nm and the wall thickness of the nanotubes is 9–10 nm, which is equivalent to 6–7 **MOC**s H-bonded with ammonium cations (Fig. 2j and Supplementary Figure 19). To get an insight into the growth of these nanotubes, FESEM images of **MOC**/aq.$NH_3$ solution are recorded at various stages of hydrogel formation (Fig. 3a–e and

Supplementary Figures 20, 21 and 22). Higher concentration of $NH_4^+$ at the early stage of gelation leads to fast aggregation of **MOC**s and results in irregular crumpled sheets, as observed from FESEM images after 2 h (Fig. 3a and Supplementary Figure 20). Each crumpled sheet acts as a nodal point for further anisotropic growth towards the closely spaced 1D tapes which are observed in FESEM images after 4 h (Fig. 3b and Supplementary Figure 21). Such anisotropic growth to 1D tape is probably governed by the competitive binding of DMA and $NH_4^+$ cations[45–47]. As the reaction proceeds, these 1D tapes are further assembled to form partially grown nanotubes in which three sides of nanotubes are

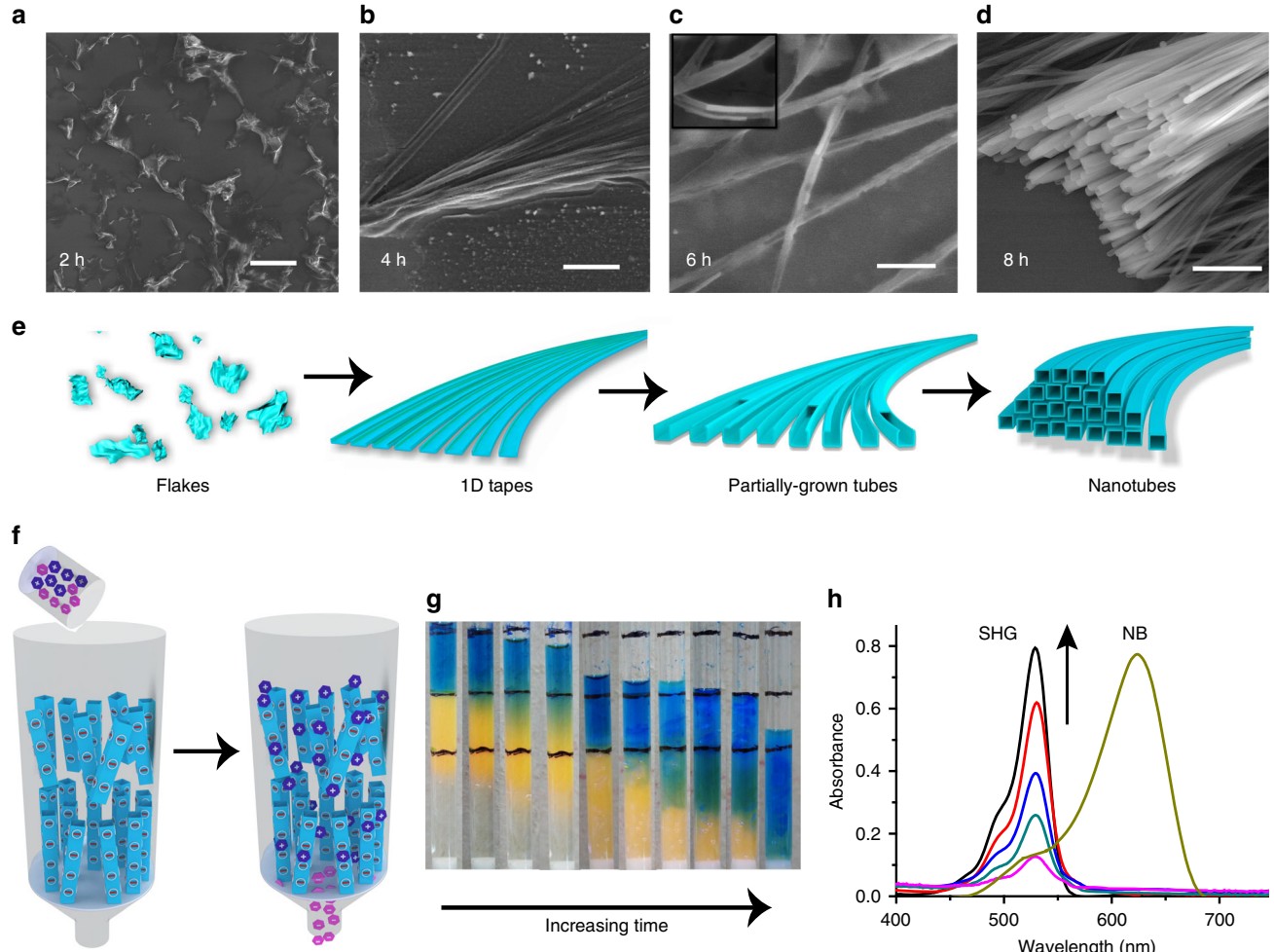

**Fig. 3** Time dependent growth of **MOC**-G1 nanotubes and gel column-chromatographic separation of charged species. **a**, **b**, **c**, **d** FESEM images of **MOC**/aq. NH$_3$ mixture taken at different time intervals during the process of gelation. **a** Scale bar = 50 μm. **b** Scale bar = 30 μm. **c** Scale bar = 1 μm and **d** scale bar = 2 μm. **e** Schematic showing time dependent change in morphology of **MOC**/aq. NH$_3$ mixture from flakes (at 2 h) to elongated tapes (at 4 h) to partially formed nanotubes (at 6 h) to complete nanotubes (at 8 h). **f** Schematic showing column-chromatographic separation of the mixture of cationic and anionic dye molecules using **MOC**-G1 hydrogel nanotubes as stationary phase. **g** Change in the colour of gel-column with time indicating separation of SHG and NB from their mixture. **h** Comparison of absorption spectra of eluent indicating gradual separation of SHG from NB. With increasing time, more SHG molecules come out of the column which is evident from increase in absorbance of 530 nm band. After all the SHG has come out, the gel-column is transformed to powder by changing pH and washed with methanol to recover NB

formed, as seen from FESEM after 6 h (Fig. 3c and Supplementary Figure 22). It is interesting to note that the solution converts to viscous liquid at this stage. After 8 h, stable transparent gel is formed and FESEM images show complete formation of nanotubes (Fig. 3d).

To generalize the hypothesis that externally added molecular binder with H-donor sites are facilitating the self-assembly of **MOC**s, we check the gelation propensity of **MOC**s in presence of various aliphatic amines and amino acids, such as N-(2-aminoethyl)-1,3-propanediamine (AEPD), guanidine hydrochloride (gua.HCl) and β-alanine (β-ala). When certain concentrations of aq. AEPD, aq. gua.HCl and aq.β-ala are added to the aqueous solution of **MOC**, within few hours the mixtures transform into stable **MOC**-G2, **MOC**-G3 and **MOC**-G4 hydrogels, respectively (Fig. 4a). FESEM and TEM images of **MOC**-G2 xerogel show bouquet-like morphology in which individual needles are several micrometer long with an approximate diameter of 200–300 nm (Fig 4b, c and Supplementary Figures 23, 24). FESEM and TEM images of **MOC**-G3 xerogel reveal sheet-like nanostructure (Fig. 4d, e and Supplementary Figure 25). Similar analysis with

**MOC**-G4 xerogel reveals nanocube morphologies with dimension in 200–250 nm range (Fig. 4f, g and Supplementary Figure 26). The above observations indicate that structure, geometry and number of H-bonding donor sites of molecular binders have immense impact on the morphology of hydrogel nanostructures. To know any specificity towards one binder, we check gelation propensity of the **MOC** towards mixture of binders (See Supplementary Method). The possible combinations are 1) NH$_4^+$ + AEPD + gua.HCl, 2) AEPD + gua.HCl + β-ala, 3) NH$_4^+$ + gua.HCl + β-ala, 4) NH$_4^+$ + AEPD + β-ala. In all cases, stable hydrogels are formed and they are named as **MOC**-GM1, **MOC**-GM2, **MOC**-GM3 and **MOC**-GM4, respectively (Supplementary Figure 27). The FESEM images of all xerogels display co-existence of three different nano-morphologies (e.g. nanotube, nano-needle and nano-sheet, for **MOC**-GM1), which indicates that the mixture of three molecular binders (e.g. NH$_4^+$ + AEPD + gua.HCl) individually drive the self-assembly of **MOC**s to respective nanostructures (Supplementary Figures 28–31). However, the distribution of three different nano-structures depends upon binding affinity of molecular binders to **MOC** (Supplementary

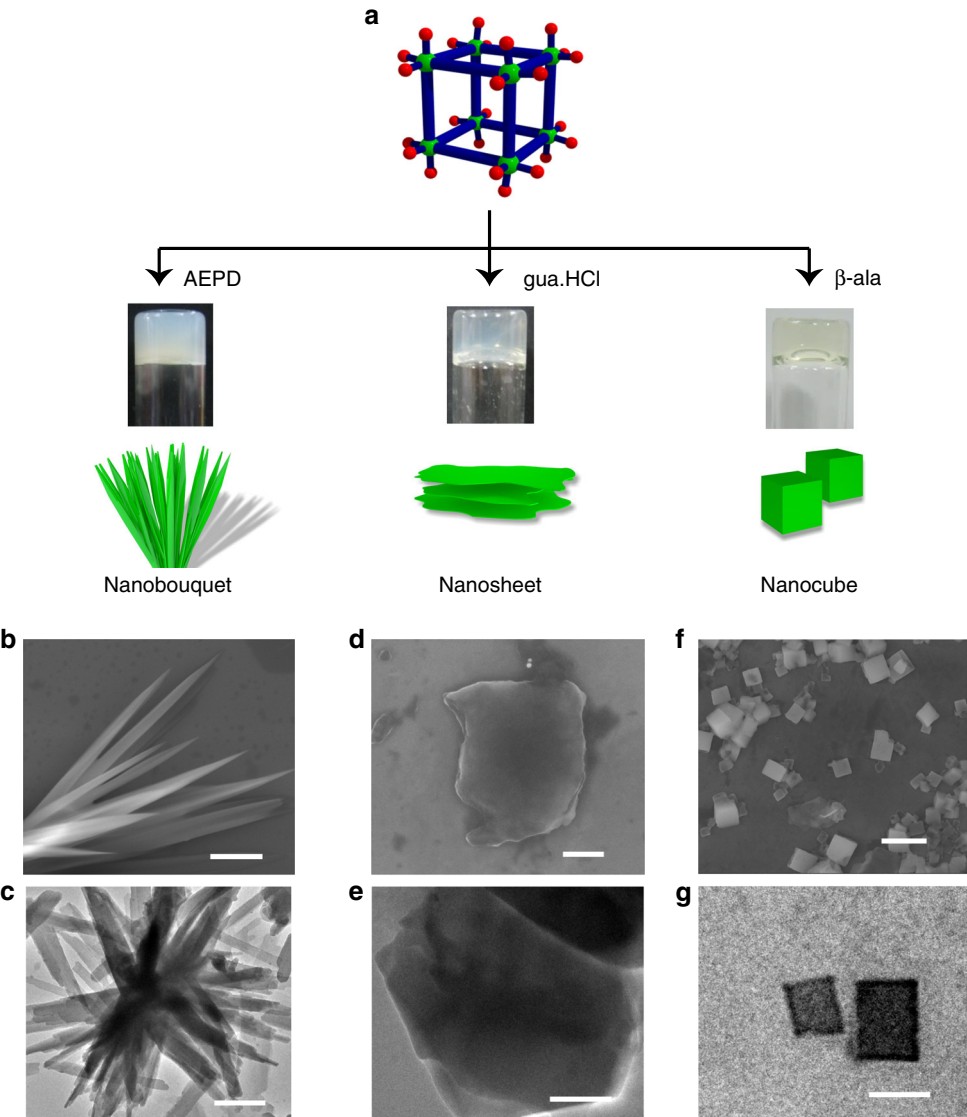

**Fig. 4** Different morphologies of **MOC**-based hydrogels. **a** Self-assembly of **MOC** with molecular binders, N-(2-aminoethyl)-1,3-propanediamine (AEPD), guanidine hydrochloride (gua.HCl) and β-alanine (β-ala) results in stable hydrogels with nano-bouquet, nano-sheet and nano-cube morphologies, respectively. **b** FESEM and **c** TEM images (scale bar = 2 μm) of **MOC-G2** hydrogel showing the formation of nano-bouquet morphology. **d** FESEM (scale bar = 500 nm) and **e** TEM (scale bar = 200 nm) images of **MOC-G3** hydrogel showing the formation of nano-sheet morphology. **f** FESEM (scale bar = 1 μm) and **g** TEM (scale bar = 200 nm) images of **MOC-G4** hydrogel showing the formation of nanocube morphology

Table 3). To further determine binding affinities of the molecular binders, isothermal titration calorimetric (ITC) experiments are performed at 25 °C. With incremental addition of $NH_4^+$, AEPD, β-ala and gua.HCl aliquots to the aqueous solution of **MOC**, exothermic heat changes are observed (Supplementary Figures 32 and 33). The binding affinity values ($K_a$) for the $NH_4^+$, AEPD, β-ala and gua.HCl binders are found to be $6.65 \times 10^4$, $6.76 \times 10^4$, $6.50 \times 10^4$ and $2.73 \times 10^4\,M^{-1}$, respectively, indicating binding affinities vary as AEPD > $NH_4^+$ > β-ala > gua.HCl (Supplementary Table 3). The results are in good agreement with the morphological distributions obtained from **MOC-GM1**, **MOC-GM2**, **MOC-GM3** and **MOC-GM4**.

**Gel-column chromatographic separation of charged species.** Zeta potential **MOC-G1** xerogel is found to be −22 mV, indicating negatively charged surfaces of the nanotube (Supplementary Figure 34). The negatively charged surface of one dimensionally aligned nanotubes prompted us to use **MOC-G1**

for gel chromatographic separation of oppositely charged species from their mixture (Fig. 3f). For this study, we have used anionic (sulforhodamine G, SHG) and cationic (nile blue, NB and acridne orange, AO) dye molecules, since they would give better visual demonstration (Supplementary Figure 35). Initial studies indicate that the hydrogel can fully adsorb (~100%) the cationic dyes ($10^{-5}$ M solution) but not the anionic dyes ($10^{-5}$ M solution) as observed from UV-Vis spectra (Supplementary Figures 36–38). For demonstrating separation of dyes from their mixture, a chromatographic column (2.5 cm long) is packed with hydrogel (stationary phase) and eluted with mixture of NB ($1 \times 10^{-6}$ M) and SHG ($1 \times 10^{-6}$ M) in methanol (Fig. 3f, g). On passing the feed solution the column first becomes orange because SHG elutes rapidly through the column due to electrostatic repulsion with nanotubes (Fig. 3g). Gradually, all SHG molecules pass through the column and at that point the column turns completely blue indicating the adsorption of NB in hydrogel (Fig. 3g). Such selective adsorption of NB in **MOC-G1** is resulting due to the electrostatic interaction of the cationic dye onto the surface of

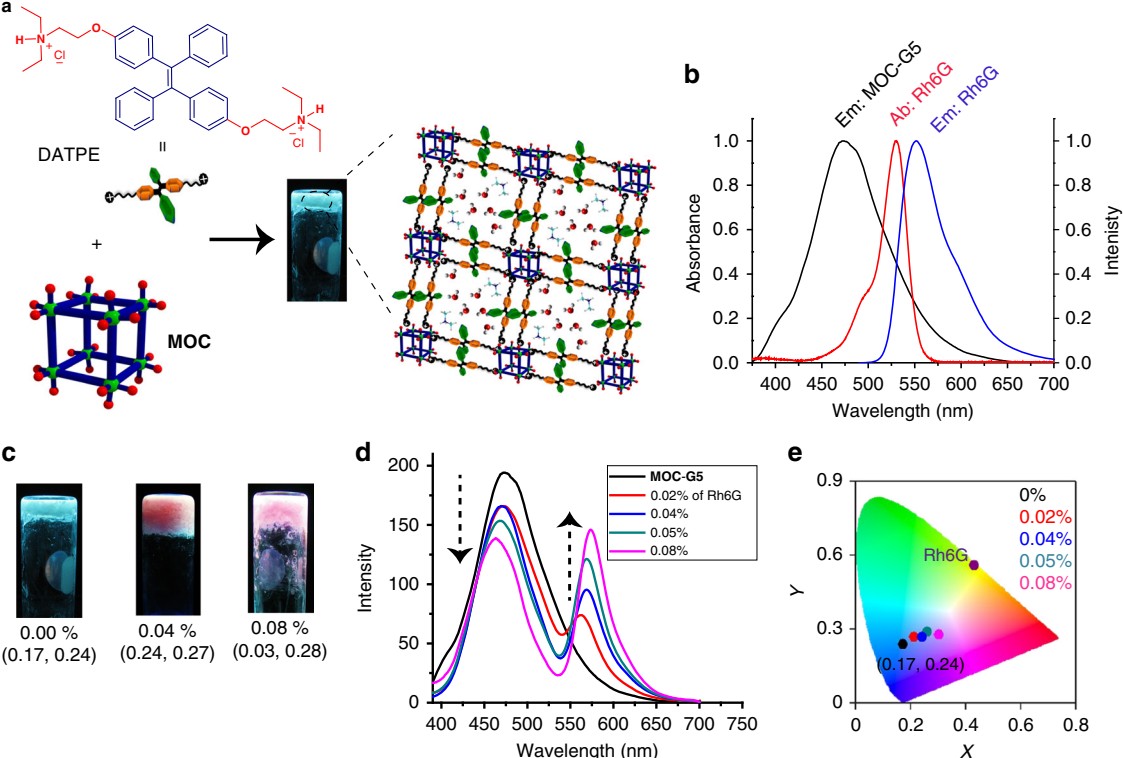

**Fig. 5** Luminescent **MOC**-**G5** hydrogel as an energy transfer scaffold. **a** Schematic showing self-assembly of **MOC** and DATPE to **MOC**-**G5** hydrogel which shows blue emission under UV light. **b** Emission spectrum of **MOC**-**G5** (black), absorption (red) and emission (blue) spectrum of Rh6G in solution. **c** Images of **MOC**-**G5** hydrogels with various loading of Rh6G (images are taken under UV lamp) and their corresponding CIE coordinates (mentioned below). **d** Emission spectra of **MOC**-**G5** hydrogels loaded with different amounts of Rh6G (arrows indicate changes) and **e** corresponding chromaticity diagram showing the CIE coordinates of the hybrid hydrogels

anionic nanotubes. Interestingly, the **MOC** does not degrade after adsorption of NB over **MOC**-**G1**. The HRMS analysis NB@**MOC**-**G1** shows peaks at $m/z = 821.58$ $(z = 3^-)$ and $1233.34$ $(z = 2^-)$ corresponding to $((Ga_8(ImDC)_{12})(2Na^+)(7\ H^+)(H_2O))^{3-}$ and $((Ga_8(ImDC)_{12})(2Na^+)(8\ H^+)(H_2O))^{2-}$ moieties, respectively (Supplementary Figure 39). Furthermore, TEM images of NB@**MOC**-**G1** show similar nanotube morphology as that of **MOC**-**G1**, suggesting NB is adsorbed on the surface of the nanotubes and does not change the morphology of **MOC**-**G1** (Supplementary Figure 40). After dye separation the hydrogel column (NB@**MOC**-**G1**) is repeatedly washed with MeOH to remove absorbed **NB** and the recovered hydrogel still exhibits nanotube morphology as confirmed by TEM images (Supplementary Figure 41). The as-synthesized **1**, which also has negatively charged surface as confirmed by zeta potential, shows minimal adsorption (29%) of NB (Supplementary Figure 42). This observation indicates that not only the negative surface but also the entangled nanotube morphology and the entrapped water molecules between them are helping the selective adsorption of cationic dyes and the fast removal of anionic dyes through gel. Unlike size selective separation broadly used in gel-chromatographic technique, here **MOC**-**G1** hydrogel is utilized for the separation of oppositely charged species based on electrostatic interactions. The pH-responsive behaviour of **MOC**-**G1** hydrogel is utilized to recycle the gel-column. After addition of few drops of 0.1 N HCl to NB@**MOC**-**G1** a precipitate forms which is washed with MeOH. Interestingly, FESEM images of the precipitate show nanocube morphology (Supplementary Figure 43) and PXRD shows peaks similar to the characteristic peaks of as-synthesized **MOC**, indicating stability of **MOC** after precipitation (Supplementary Figure 44). The **MOC** precipitate can

easily be converted to **MOC**-**G1** hydrogel after addition of aq. $NH_3$ and hydrogel column can be regenerated. Next, we choose rhodamine B (RhB) dye which have both cationic group $((Et)_2N^+)$ and the easily deprotonable carboxylic acid group $(-COOH)$. The hydrogel adsorbs only 34% RhB (Supplementary Figure 45). Since the hydrogel is basic (pH = 11) the $-COOH$ group of RhB gets deprotonated easily while passing through the gel and hence is repelled by the surface of anionic nanotubes. On the other hand, the cationic part of RhB $((Et)_2N^+)$ force the molecule to be attached with the gel matrix. Because of such two opposite forces, only 34% dye is absorbed by **MOC**-**G1**.

**Preparation of luminescent MOC-G5 hydrogel**. A cationic, AIE-active molecular binder, 1,2-bis(4-(2-diethylammonioethoxy)phenyl)-1,2-diphenylethenedihydrochloride (DATPE) having flexible ethoxy chain functionalized with terminal quaternary ammonium $(REt_2NH^+)$ groups is designed to achieve CAHB assisted assembly of **MOC**s (Fig. 5a). It is envisioned that assembly of DATPE and **MOC**s will result in highly emissive hydrogel due to restricted phenyl rotation of TPE segments. The dilute solution of DATPE shows weak or no emission while the concentrated solution shows strong AIE emission at 474 nm $(\lambda_{ex} = 350\ nm)$. Initially, the interaction of **MOC** and DATPE in solution is studied by gradually adding equivalents of DATPE into **MOC** (10 μM) solution (Supplementary Figure 46). The **MOC** solution which is otherwise non-emissive shows a band at 474 nm $(\lambda_{ex} = 350\ nm)$ after addition of 0.8 eq of DATPE. With incremental addition of DATPE the emission intensity at 474 nm increases and saturates after addition of 6 eq DATPE. This strongly suggests that DATPE indeed interacts with **MOC**s in

solution, feasibly through charge assisted hydrogen bonding. Interestingly, formation of cyan emissive aggregates is observed at this point (Supplementary Figure 46). Such strong emission of aggregates is attributed to the combined effect of immobilization and aggregation induced restriction of phenyl rotation of DATPE. Room temperature mixing of aq. **MOC** (12 mM) and aq. DATPE (8.3 mM) (0.3 eq of DATPE) followed by sonication result in stable hydrogel **MOC-G5** (Fig. 5a). Rheology study of **MOC-G5** shows that $G'$ is higher than the loss modulus $G''$ and both of them ($G'$ and $G''$) are independent of $\omega$ over a large range of strain (%) indicating viscoelastic nature of **MOC-G5** (Supplementary Figure 47). The value of $G'$ at high strain (%) is 11520 Pa. The stress versus strain plot shows the yield stress and yield strain values as 163 Pa and 3.72%, respectively (Supplementary Figure 47). PXRD pattern of **MOC-G5** xerogel shows the presence of Bragg's reflections as observed in **1**, suggesting **MOC**s are intact in **MOC-G5** xerogel (Supplementary Figure 48). FTIR of xerogel shows characteristic peaks of **MOC** as well as DATPE, suggesting their existence in the hybrid gel (Supplementary Figure 49). FESEM and TEM images of the xerogel reveal presence of cubic particles with size in the range of 500–800 nm (Supplementary Figure 50). The self-assembly of **MOC** and DATPE is explained by a proposed mechanism, shown in Fig. 5a. Both titration study and molecular dimension of DATPE (size: $19.0 \times 7.0$ Å$^2$) suggest that two cubes can be connected to maximum two DATPE molecules in a face-to-face manner via CAHB interaction between free oxygen atoms of **MOC** and amine groups of DATPE (Supplementary Figures 51–52). Therefore, each **MOC** would be attached to six neighbouring **MOC**s by 12 DATPE molecules and forms smaller aggregate which extend three dimensionally to generate a network like structure entrapping DMA cations and water molecules to form hydrogel (Fig. 5a). **MOC-G5** also shows strong cyan emission at 474 nm ($\lambda_{ex} = 350$ nm) similar to molecular aggregates formed during titration study, with a CIE coordinates of (0.17, 0.24) (Fig. 5b, c). The restricted phenyl rotation in gel initially occurs through immobilization of DATPE over **MOC** followed by aggregation effect, which is due to $\pi$–$\pi$ interactions between two neighbouring DATPE moieties. Therefore, the emission in **MOC-G5** is a combination of immobilization and AIE phenomena as previously observed in TPE-based MOFs and coordination polymer gels[48,49].

**Photomodulated emission in MOC-G5**. TPE shows strong AIE emission in solid state and it is well-known to undergo photocyclization to diphenylphenanthrene (DPP), which is non-emissive in solid state and blue emissive in solution (Fig. 6a and Supplementary Figure 53)[50,51]. This prompt us to study the photo-responsive modulation of emission in **MOC-G5** hydrogel. We initially test photocyclization of pure DATPE dissolved in water ($D_2O$ for $^1$H-NMR study). Aqueous solution of DATPE shows weak emission at 470 nm ($\lambda_{ex} = 350$ nm) (Supplementary Figure 54). Upon photo-irradiating the solution with 365 nm light source (22.5 W) for 15 min, a weak shoulder at 410 nm appears and its intensity gradually increases with increasing the irradiation time. This new peak corresponds to the photocyclized DATPE (to DPPQA). After 45 min of irradiation the peak at 470 nm totally diminishes and only the DPPQA emission at 410 nm is observed (Supplementary Figure 54). The changes can also be observed by naked eye as emission colour of the solution under UV light completely changes from weakly emissive to highly blue emissive (Supplementary Figure 55). This result is consistent with the literature[50,51]. To further support the photocyclization, $^1$H-NMR spectrum of DATPE in $D_2O$ is recorded at different time intervals. The peaks at 6.38 and 6.45 ppm gradually diminish and new peaks arise at 7.47, 7.60, 8.01, 8.25 and 8.68 ppm

(corresponding to DPPQA), confirming photocyclization of DATPE to DPPQA (Supplementary Figure 55). With these convincing results, we further study the photocyclization of DATPE in **MOC-G5** hydrogel. The emission spectrum of **MOC-G5** gel film prepared on quartz substrate exhibits strong AIE emission peak at 474 nm due to DATPE (Supplementary Figure 56). However, after photo-irradiating (with 365 nm light) the film for 20 min the emission gets completely quenched, which is also clear from images of the film under UV lamp (Supplementary Figure 56). This quenching of emission is attributed to the aggregation caused quenching (ACQ) effect of DPPQA formed after photo-cyclization of DATPE in the film. However, on dissolving this photo-irradiated film in water a strong blue emission at 410 nm is observed indicating the presence of molecularly dissolved DPPQA units (Supplementary Figure 57). It is indeed possible to photopattern/write the desired shapes/letters with this hydrogel. To elucidate this, a **MOC-G5** film is prepared on glass substrate and a light opaque template with unmasked pattern of 'J N C' is made. The opaque template is placed over the **MOC-G5** glass substrate. Upon irradiation ($\lambda = 365$ nm) for about 20 min the unmasked area turns to non-luminescent due to formation of DPPQA (ACQ effect) while the masked area remains luminescent (Fig. 6e).

**Processable white-light-emitting hydrogel**. White light emission can be achieved by balancing the ratio of cyan and orange or RGB colours[52–54]. Here, a dichromatic approach is utilized to achieve white-light-emitting **MOC** hybrid gel. As mentioned earlier, **MOC-G5** showed strong AIE emission at 474 nm ($\lambda_{ex} = 350$ nm) with a CIE coordinates of (0.17, 0.24) (Fig. 5c, e). We envision that such luminescent hybrid gel can act as donor scaffold for partial energy transfer to immobilized acceptor dye and thus could exhibit tunable emission property. Rhodamine 6 G (Rh6G) is chosen as an acceptor because its absorption spectrum overlaps partially with emission spectrum of **MOC-G5** hybrid and also its cationic nature would help immobilization over the hydrogel matrix (Fig. 5b). In-situ mixing of Rh6G (0.02 mol%) during gelation under sonication triggers instant formation of a tri-component hydrogel, **Rh6G$_{0.02\%}$@MOC-G5**. Interestingly, the emission of **Rh6G$_{0.02\%}$@MOC-G5** at 474 nm ($\lambda_{ex} = 350$ nm) decreases to an extent while a new band appears at 560 nm from encapsulated Rh6G (Fig. 5d). With gradual increase of Rh6G concentration from 0.02 to 0.08%, the emission intensity at 474 nm exhibits a gradual decrease with an simultaneous increase in intensity of Rh6G band (Fig. 5d). Interestingly, the emission colour of hydrogels changes from cyan to strong pink when observed under UV light (Fig. 5c). CIE coordinates of **Rh6G$_{0.08\%}$@MOC-G5** are calculated to be (0.30, 0.28), which is near white light as seen in CIE diagram (Fig. 5e). Energy transfer in the mixed hybrid is evident from direct excitation of **Rh6G$_{0.08\%}$@MOC-G5** at 530 nm which shows less emission intensity compared to the indirect excitation at 350 nm (Supplementary Figure 58). The excitation spectrum of **Rh6G$_{0.08\%}$@MOC-G5** collected at 580 nm shows maximum intensity at 350 nm confirming the contribution of DATPE to the observed emission at 580 nm (Supplementary Figure 59). Moreover, fluorescence decay profile of **Rh6G$_{0.08\%}$@MOC-G5** monitored at the DATPE emission (470 nm, $\lambda_{ex} = 350$ nm) exhibits shorter lifetime values (2.4 ns) compared to **MOC-G5** hybrid gel (4.5 ns), clearly suggesting efficient excitation energy transfer through Förster resonance energy transfer (FRET) mechanism (Supplementary Figure 60). The energy transfer efficiency ($\Phi_e$) and rate constant ($k_e$) of **Rh6G$_{0.08\%}$@MOC-G5** are calculated to be 45.8% and $1.876 \times 10^8$ s$^{-1}$, respectively (See Supplementary Discussion). It is expected that cationic Rh6G molecules interact

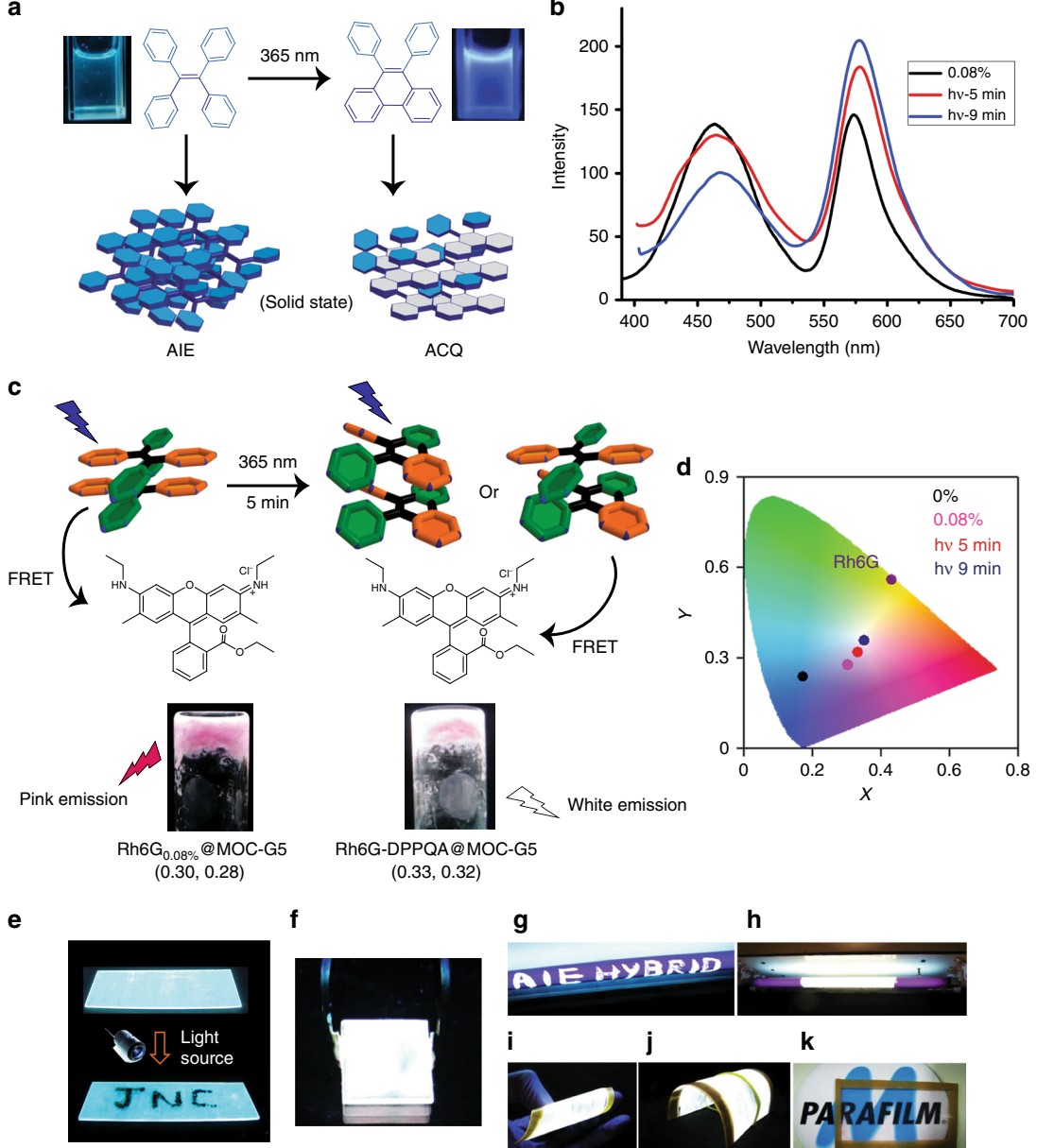

**Fig. 6** Photomodulated white light emission in hydrogel. **a** A general schematic showing photocyclization of TPE (tetraphenylethene) to DPP (diphenylphenanthrene) on irradiation with 365 nm light. **b** The emission spectrum of **Rh6G_{0.08%}@MOC-G5** before photoirradiation (black), after 5 min (red) and 9 min (blue) irradiation with 365 nm light source. **c** Schematic showing the energy transfer pathways before and after photoirradiation. The corresponding CIE coordinates of **Rh6G_{0.08%}@MOC-G5** and white-light-emitting gel (**Rh6G-DPPQA@MOC-G5**) are mentioned below the respective hydrogel images. **d** Chromaticity diagram showing corresponding CIE coordinates of **MOC-G5** (black), **Rh6G_{0.08%}@MOC-G5** (pink), white-light-emitting hydrogel (red, form after 5 min of photoirradiation on **Rh6G_{0.08%}@MOC-G5**), hydrogel (navy blue, form after 9 min of photoirradiation on **Rh6G_{0.08%}@MOC-G5**) and Rh6G (wine). **e** A glass slide coated with **MOC-G5** hydrogel is irradiated with 365 nm light to write 'JNC'. Processability of white-light-emitting hydrogel: **f** image of glass substrate coated with white-light-emitting hydrogel under UV lamp, **g** writing and **h** painting using same hydrogel on 365 nm UV lamp. **i**, **j** Images of a flexible plastic substrate (5 × 3 cm) coated with white-light-emitting-hydrogel under UV light and **k** under day light

with anioinc **MOCs** electrostatically and their appropriate dipole-dipole orientation with DATPE molecules in gel system results in energy transfer through FRET mechanism. Further increasing the Rh6G loading to 0.1% yields a viscous liquid and no gel formation is observed even after 24 h.

The emission feature of **Rh6G_{0.08%}@MOC-G5** gel with CIE coordinates (0.30 0.28) is further tuned towards white-light-emission by increasing the contribution of Rh6G emission. To do this, photocyclization property of TPE segment is utilized (Fig. 6a). We envisioned that the controlled photo-irradiation on **Rh6G_{0.08%}@MOC-G5** could lead to the formation of DPPQA

in hydrogel and the contribution of blue emitting DPPQA molecules would further broaden the emission spectrum of photo-irradiated **Rh6G_{0.08%}@MOC-G5** (Fig. 6b, c). Interestingly, photo-irradiation of **Rh6G_{0.08%}@MOC-G5** hydrogel for 5 min appreciably broadens the 474 nm band which increases the spectral overlap of donor emission with acceptor (Rh6G) absorption and eventually an increase in extent of energy transfer is observed (Fig. 6b). Excitation of photoirradiated (5 min) **Rh6G_{0.08%}@MOC-G5** at 350 nm indeed shows an enhancement in the emission intensity at 580 nm due to enhanced energy transfer efficiency (Fig. 6b). Remarkably, emission colour of the

gel changes from pink to white as seen under UV lamp. The CIE coordinates of photo-irradiated gel (**Rh6G-DPPQA@MOC-G5**) are found to be (0.33, 0.32) indicating a pure white-light-emission from the hybrid gel (Fig. 6d). Positive ion acquisition mode HRMS analysis of the photo-irradiated **Rh6G$_{0.08\%}$@MOC-G5** shows nearly 8% conversions of DATPE molecules to DPPQA on 5 min photoirradiation (Supplementary Figure 61). The stability of **MOC** in the photo-irradiated **Rh6G$_{0.08\%}$@MOC-G5** is confirmed by negative ion acquisition mode HRMS analysis which shows peak at $m/z = 1233.26$ ($z = 2^-$), corresponding to $((Ga_8(ImDC)_{12})(2Na^+)(8\,H^+)(H_2O))^{2-}$ moiety (Supplementary Figure 62). The nanocube morphology is retained in **Rh6G$_{0.08\%}$@MOC-G5** after photo-irradiation (Supplementary Figure 63). Energy transfer in white-light-emitting hybrid is further evident from direct excitation of the Rh6G at 530 nm, which shows less emission intensity compared to the indirect excitation at 350 nm, clearly suggesting efficient excitation energy transfer through FRET mechanism (Supplementary Figure 64). Energy transfer is further evident from excitation spectrum of photoirradiated **Rh6G$_{0.08\%}$@MOC-G5** collected at 580 nm, which shows maximum intensity at 350 nm confirming the contribution of DATPE to the observed emission at 580 nm (Supplementary Figure 65). Fluorescence decay profiles of white-light-emitting gel monitored at DATPE emission (474 nm) shows shorter life time value (1.5 ns) than **MOC-G5** hybrid gel (4.5 ns) clearly indicating excitation energy transfer from DATPE to Rh6G in the hybrid gel (Supplementary Figure 66). The energy transfer efficiency ($\Phi_e$) and rate constant ($k_e$) are calculated to be 66.7% and $4.446 \times 10^8\,s^{-1}$, respectively in white-light-emitting hydrogel. This prove that photo-irradiation indeed enhances the energy transfer efficiency. The absolute quantum yield of white-light-emitting gel is found to be 16%.

Easy reversible sol-gel transformation of white-light-emitting hydrogel prompted us to study large area coating for device fabrication (Fig. 6f). This hybrid white-light-emitting hydrogel can be easily painted on a commercially available UV lamp (Fig. 6g, h). Upon turning on the UV lamp the uncoated portion remains dull blue while coated part strongly illuminates white light. Similarly, letters written on the UV lamp with the soft-hybrid become readable when power source is on, as they emit the white light upon excitation. Also, the hybrid hydrogel can be coated on flexible plastic substrates that not only exhibits strong white emission under UV lamp but also remain highly transparent under day light (Fig. 6i–k).

## Discussion

In summary, we have prepared water soluble, Ga$^{3+}$ bases anionic **MOC** which self-assemble to hydrogel in presence of different molecular binders through charge-assisted H-bonding interaction. Depending upon shape and geometry of the molecular binders, the hydrogels show different morphologies, such as nanotube, nano- bouquet, nanosheet and nanocube. Moreover, the properties of the hydrogels are tuned by selecting the suitable binders. Here we have exploited two different properties of the **MOC**-based hydrogels. In one hand, the surface negative charge of the nanotubes of **MOC-G1** is exploited for gel-chromatographic separation of cationic dyes from anionic dye. On the other hand, we have documented a photo-responsive luminescent hydrogel based on AIE active chromophoric binder that is further exploited for light harvesting application. In addition, we also prepared a white-light-emitting hydrogel by tuning the donor-acceptor energy transfer efficiency using photo-cyclization of TPE as a tool. In short, **MOC**-hydrogels provide a platform to integrate different type of molecular binders with

**MOC** to form self-assembled nanostructures, where the properties and corresponding functionalities can be deliberately tuned.

## Methods

**Synthesis of $((Me_2NH_2)_{12}(Ga_8(ImDC)_{12})\cdot DMF\cdot 29H_2O)$ (1).** H$_3$ImDC (0.5 mmol, 78 mg), Ga(NO$_3$)$_3$.H$_2$O (0.5 mmol, 128 mg) and 10 ml DMF were mixed in a 20 ml Teflon container and stirred for 30 minutes at room temperature. A volume of 15 μl NEt$_3$ was added into the reaction mixture and stirred for additional 30 minute. After that the Teflon container was kept inside a stainless steel autoclave which was heated at 120 °C for 24 h. After the reaction was over, the autoclave was slowly cooled down to room temperature. The white product formed was centrifuged and washed repeatedly by methanol and dried in air. The air dried powder was dissolved in 10 ml water and kept for recrystallization at room temperature. Within 4 days colourless, block shaped crystals were formed. The crystal structure determination reveals the molecular formula of the compound as $((Me_2NH_2)_{12}(Ga_8(ImDC)_{12})\cdot DMF\cdot 29H_2O)$. Selected FTIR data (KBr, cm$^{-1}$): 3447 (b), 3086 (m), 2775 (m), 1676 (s), 1473 (s), 1363 (s), 1100 (s), 857 (m), 660 (m), 550 (m). Anal. Calcd for C$_{93}$H$_{169}$Ga$_8$N$_{39}$O$_{80}$: C, 29.53; H, 4.93; N, 14.65. Found: C, 29.14; H, 4.55; N, 14.76. The phase purity of the powder sample was checked by comparing PXRD of the bulk powder sample with the simulated data from single crystal.

**Synthesis of DATPE.** The synthesis and characterization of DATPE have been shown as Supplementary Methods in Synthesis and Characterization.

**Preparation of MOC-hydrogels.** For preparing **MOC-G1** hydrogel, 15 mg **1** was dissolved in 1 ml water and 100 μl aq NH$_3$ is added into the solution. The mixture was sonicated for few minutes and kept at room temperature. **MOC-G1** hydrogel was formed after 8 h. For preparing **MOC-G2** hydrogel, 20 mg **1** was dissolved in 500 μl water and 500 μl N-(2-aminoethyl)-1,3-propanediamine solution (0.126 M) was added dropwise into it. The mixture was sonicated for few minutes and kept undisturbed at room temperature. The mixture became viscous after 4–5 h and formed stable **MOC-G2** hydrogel after one day. For preparing **MOC-G3** hydrogel, 20 mg **1** was dissolved in 300 μl water and 600 μl solution of guanidine hydrochloride (0.1 M) was added into the solution. The mixture was sonicated for few minutes and kept undisturbed at room temperature for one day to form **MOC-G3** hydrogel. For preparing **MOC-G4** hydrogel, 20 mg **1** was dissolved in 500 μl water and 500 μl solution of β-alanine (0.1 M) was added into the solution. The mixture was sonicated for few minutes and kept undisturbed. The mixture converted to stable gel after 1 day. For preparing **MOC-G5** hydrogel, 12 mg **1** was dissolved in 500 μl water. 30 μl DATPE was dissolved in 500 μl water and the solution was drop-wise added into the solution of **1**. The mixture was sonicated for few minutes. An opaque gel was formed instantaneously. In all cases formation of hydrogels was confirmed by inversion test method. More detail synthesis and characterization of **MOC**-based hydrogels containing mixture of binders have been discussed in Supplementary Methods.

**Preparation of hydrogel column for separation.** A glass column (diameter = 0.5 cm and length = 10 cm) having silica base was packed with **MOC-G1** hydrogel to prepare the gel-column (length of the stationary phase = 2.5 cm). The mixture of NB ($1 \times 10^{-6}$ M) and SHG ($1 \times 10^{-6}$ M) in methanol was layered on the hydrogel column and eluted through the gel-column. The solution coming out from the column was collected in a vial and monitored by UV-Vis spectra.

## Data availability

The X-ray crystallographic data that support the finding of this study are available in Cambridge Crystallographic Data Centre (CCDC) under deposition number CCDC 1587189. All other data supporting the findings are available in the article as well as the supplementary information files and from the authors on reasonable request.

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

## Acknowledgements

P.S and V.M.S are thankful to Council of Scientific and Industrial Research (CSIR), GOI for fellowship. T.K.M is grateful to the SERB, Department of Science and Technology (DST, Project No. MR-2015/001019), Government of India (GOI) and JNCASR for financial supports. The authors acknowledge Dr. S. J. George and Mr. S. Kuila for lifetime measurements; Dr. R. Ganapathi for rheology measurements; Prof. T. K. Kundu and Dr. S. Kaypee for helping in ITC measurements.

## Author contributions

P.S., V.M.S. and T.K.M. designed the concept of this work. P.S. and V.M.S. contributed equally in carrying out all the experimental works. K.J. and A.H. contributed in solving the crystal structure of **1**. P.S., V.M.S. and T.K.M. analysed the experimental data and

wrote the manuscript. All the authors discussed over the results and commented on the manuscript.

## Additional information

**Competing interests:** The authors declare no competing interests.

