## [Peer Review File · Nature Communications]

Reviewers' comments:

Reviewer #1 (Remarks to the Author):

This manuscript reports the formation of a metal organic cube $\{(Me_2NH_2)_{12}[Ga_8(ImDC)_{12}] \cdot DMF \cdot 29H_2O\}$, which can then be transformed into a gel, after dissolution in water and addition of one of several organic cations – the nature of which controls the gel morphology.

The work is well referenced, though the novelty is highly specific and the broad idea of linked metal organic cubes has been explored previously, in the works cited by the author, e.g. Refs 40-42. It may be argued that this current work is very similar to reference 42, though with a much shorter cube spacer.

I find the manuscript at present to be too dense to of interest to a non-specialist audience. The figures would be of particular concern to me. In general, they add little information to guide the author, are crowded and repeat information. Specific examples are as follow:

I could not find Figure 1 mentioned in the text, and I do not see how it fits in given the redundancy of information. Figure 1a displays much of the same information as 2f and 2g, and the same optical image is in 1a and 2e. Figure 2 is extremely crowded. The references to figures in the text all need thoroughly checking as I believe many to be incorrect – “The above analysis indicates that the growth mechanism involves nucleation, followed by aggregation of particle seed, fusion, growth and finally annealing processes ⁴⁵⁻⁴⁷ (Fig. 2e).” and “long tubular nanostructures with rectangular cross-section (Fig. 2g) as examples.

The manuscript would be substantially improved if, after significant clarification, the authors were able to put these materials into context with others regarding the tested applications of gel-chromatographic separation and flexible light emitting displays. At present, it is unclear to the reader what the specific advantages are.

The authors might also comment further on links to metal organic framework gels, such as DOI:10.1038/nmat5050, and give an indication of how thermal and mechanical stability compares to other materials once dried.

More minor points are listed below

- Abstract: Several small omissions. “Well defined metal-organic cubes (MOC) into”. “Like the”
- I disagree with the authors assertion that “PXRD of 1 and xerogel showed no significant change, which suggests MOCs are intact”. Peaks have appeared/disappeared, shifted position and got stronger/reduced in intensity all over here....
- The SI needs to be corrected. There are several figures that are separated from their titles, which is perhaps a PDF conversion problem. The capture for Fig. S2 should read ready cubes. The figure label for S10 should, perhaps, read minutes, not mints.

Reviewer #2 (Remarks to the Author):

The manuscript entitled "Binder driven self-assembly of metal-organic cubes towards functional hydrogels" demonstrates formation of hydrogels from metal-organic cube by using molecular binders. The authors cleverly showed interesting applications for the hydrogels produced via assembly, for example gel-chromatographic separation of cationic species from anionic counterparts, tuning emission color etc. The manuscript is of high quality in terms of experimental procedure and applications. However the reviewer would like to mention a few points and seeking clarifications for others:

1. The reviewer suggests including a detailed explanation of confirmation of intact MOCs in solution.
2. What is the effect of temperature on MOC-G1 hydrogel?
3. The strength of MOC-G1 hydrogel in terms of a storage modulus (G') should be given in the main text. Is it comparable with the existing hydrogels formed from metal-organic cages linked by molecular binders?
4. Rheology results of MOC-G5 should be included in supporting information.
5. Although application of the MOC-based gel for light harvesting application is novel, mixing different dyes such as rhodamine and tetraphenylethene in a gel matrix to get white light emission is already reported in literature. Several of fluorescent hydrogels are known to be used for photo patterning and white light emission. Therefore advantage of the present system over the existing methods/systems should be addressed.
6. The concept is already reported in ref. 40-42
7. Use of the material for the column-chromatographic separation oppositely charged species is a novel concept and is well demonstrated.
8. In Figure (3) caption Spelling 'cholumn' should be corrected
9. In Figure 1, Nano-tube gel and nano-cube gels are transparent while others are opaque, justify.
10. Figure 2. Pretty images rectangular tubes were shown. I am wondering why they are rectangular in shape, whether the inner tube is hollow, what is the size of the inner channel (150-200 nm)? This value corresponds to the size of a cube? Please explain
11. Figure 3. Is there is any suggestive mechanism of tube growth (proposed scheme), especially from 1D tape to a hollow rectangular tube? Figure 3 b, and c are not convincing enough – it can be due to drying induced artifact too. Replace with better images, if available. The Chromatographic scheme is well done!
12. Figure 4. What drives the mechanism of growth into different morphologies with different binders? Among the three binders, is there any specificity towards one binder, if a mixture was used?

In summary, this is an interesting piece of work to create hydrogels from discrete nanocubes via self-assembly. The authors elegantly showed multiple possible applications of such gels. However, several clarifications are needed and the reviewer suggest for a major revision of the manuscript.

Reviewer #3 (Remarks to the Author):

In this manuscript, the authors have demonstrated the charge-assisted hydrogen bond driven self-assembly formation of hydrogels from a Ga^{3+} anionic metal-organic cube (MOC) and cationic binders in water. The manuscript is well written and to the point and has some very interesting results. The focus herein is on the material site and the authors observed that the morphology of resulting hydrogels is depends on the size, shape and geometry of the molecular binder used to make the gels. The nanotubular hydrogel obtained from the combination of MOC and NH_4^+ was used as stationary phase for the selective separation of cationic dyes (nile blue and acridine orange) from their anionic counterpart (sulforhodamine G). Furthermore, authors have also made

few luminescent hydrogels and employed them as light harvesting antenna to tune the emission colour to develop white light emitting devices.

The use of charge-assisted hydrogen bonding (CAHB) interaction to generate self-assembled structures and soft-materials is not something new (as the authors point out), and there are several reports of using MOC as building blocks to generate self-assembled structures and materials. Though the idea of using CAHB interaction as primary driving force to generate hydrogels is not novel, the morphological analysis and the application of hydrogels for the chromatographic separations of dyes is interesting and this is the strength of this manuscript. Overall the work described in this article will be of interest to a wide range of researchers and thus I am recommending that it can be considered for publication subject to a revision.

1. It is known that even free carboxylate ions form self-assembled nanomaterials with interesting morphological features upon binding with cationic molecular binder through charge-assisted hydrogen bonding with interactions. Therefore, I request the authors to analyse the morphological features of 4,5-imidazoledicarboxylic acid, that was used to make MOC, in presence of molecular binders. This will help the reader to understand the role of anionic metal-organic cages with confined cavities to form the self-assembled supramolecular hydrogels.

2. The stability of MOC is a series concern of the work described in this article. How stable the MOC is after the hydrogel formation and dye encapsulation and photoirradiation?

3. It is interesting to see that MOC forms hydrogel of different morphology depending on the size, shape and geometry of molecular binder. Is MOC form hydrogel in presence of more than one binder? Any change in the morphological features in presence of more than one binder at once?

4. MOC-G1 was used for gel chromatographic separation of cationic dye molecules from the anionic counterpart. MOC-G1 was regenerated after washing the HCl. The size and geometry of cationic dye is larger than the NH_4^+ . Is MOC-G1 exhibit nano-tube like structure after the adsorption and separation of dyes? How stable the MOC-G1 after washing with HCl?

5. The morphology and stability of MOC-G5 gel also need to be checked after the photoirradiation and rhodamine encapsulation.

Author reply to the reviewers' comments for the manuscript 'NCOMMS-17-33348-T'

Reviewer #1

1. The work is well referenced, though the novelty is highly specific and the broad idea of linked metal organic cubes has been explored previously, in the works cited by the author, e.g. Refs 40-42. It may be argued that this current work is very similar to reference 42, though with a much shorter cube spacer.

Response:

We are thankful to the reviewer for valuable comments. *Nat. Chem.*, **8**, 33-41 (2016) by Zhukhovitskiy *et al.* is indeed an excellent work illustrating a new type of gel, also known as polyMOC gel, assembled from polymeric ligands and metal-organic cages as junctions. In their report, when soluble polymers having coordinating end groups are mixed with appropriate metal ions, the mixture leads to the *in-situ* formation of metal-organic cage at the junction of cross-linked polymers and eventually forms polyMOC gels. The structural features of metal-organic cage were validated using indirect methods, including NMR, SANS, simulations etc. Nevertheless, the direct method such as single crystal X-ray diffraction (XRD) structural analysis of metal-organic cage is the most powerful and highly demanding analytical method to investigate internal structure. In our approach, preformed metal-organic cubes (**MOCs**) (fully characterized by single crystal method) are assembled with different small, H-bond donor molecules (molecular binders) to form hydrogels. The single crystal X-ray diffraction analysis of **MOC** provided accurate information related to the surface negative charge of **MOCs** and charge assisted H-bonding interaction with dimethyl ammonium cations (formed *in-situ* during reaction). High solubility of **MOCs** in water and its ability to form charge assisted H-bond with H-bond donor molecules prompted us to study the hydrogelation with different molecular binders including ammonium cation, N-(2-aminoethyl)-1,3-propanediamine, guanidine hydrochloride, β -alanine and chromophoric TPE-based cation. We believe our concept is applicable to different types of H-bond donating molecular binders and opens many new interesting applications. Such hydrogels formed by the self-assembly of preformed metal-organic cubes and molecular binder are yet to be accounted. We believe this water soluble **MOC** could be used as a platform to explore different types of properties of the hydrogels in future.

2. I find the manuscript at present to be too dense to of interest to a non-specialist audience. The figures would be of particular concern to me. In general, they add little information to guide the author, are crowded and repeat information. Specific examples are as follow:

I could not find Figure 1 mentioned in the text, and I do not see how it fits in given the redundancy of information. Figure 1a displays much of the same information as 2f and 2g, and the same optical image is in 1a and 2e. Figure 2 is extremely crowded. The references to figures in the text all need thoroughly checking as I believe many to be incorrect – “The above analysis indicates that the growth mechanism involves nucleation, followed by aggregation of particle seed, fusion, growth and finally annealing processes 45-47 (Fig. 2e).” and “long tubular nanostructures with rectangular cross-section (Fig. 2g) as examples.

Response:

We are thankful to the reviewer for important suggestions. We have mentioned Figure 1 in the text of revised main manuscript. Figure 1 is basically a general scheme showing the supramolecular self-assembly of **MOC** in presence of different molecular binders that result in hydrogels with different morphologies.

Figure 1a shows the schematic of self-assembly of **MOC** with binders. We have deleted the representative hydrogel image and given a hydrogel cartoon. Figure 2f shows how **MOC** would form charge-assisted H-bond with ammonium cations in **MOC-G1** hydrogel. We have deleted the Figure 2g and Figure 2 has been rearranged accordingly.

We have thoroughly checked the references of the figures in main text and corrected these mistakes in the revised main manuscript. The line “long tubular nanostructures with rectangular cross-section..” is cited with Fig. 2h-i.

3. The manuscript would be substantially improved if, after significant clarification, the authors were able to put these materials into context with others regarding the tested applications of gel-chromatographic separation and flexible light emitting displays. At present, it is unclear to the reader what the specific advantages are.

Response:

We are thankful to the reviewer for this important suggestion. Gel-chromatography is a known technique used for separating mixture of chemicals by exploiting the differences in the rates at which they pass through a bed of a porous gel matrix. In this case, the separation of the components is based on the differences in the molecular sizes of the components. Small molecules tend to diffuse into the interior of the porous particles so that their flow is restricted, while large molecules are unable to enter the pores and tend to flow through the gel-matrix. Thus, the components of highest molecular weight leave the gel-bed first, followed by successively smaller molecules. However, in our case the gel-chromatographic separation using **MOC-G1** hydrogel is based on electrostatic interaction of the components with the negatively charged surface of the nanotubes. Therefore, after gel-chromatographic

separation the cationic dye molecules get entrapped in the gel matrix and anionic dye molecules pass through the gel matrix. Hence using our technique two molecules with same size but different in charge could be easily separated from mixture.

White light emitting MOC-hybrid hydrogels offer high processability over powder solids such as MOFs and they are environment-friendly and stable for months compared to organogels. The inorganic-organic hybrid templating approach used here offers excellent organization of organic chromophores with suitable interchromophoric distances and inhibit possible aggregation caused quenching (ACQ) effects usually observed in organic chromophores. Due to easy reversible sol-gel transition, white light emitting hydrogel shows high processability and therefore easily be painted on a commercially available UV lamp and large, flexible plastic surfaces as showed in the manuscript. Further, the structural constraints in tetraphenylethene induced by anionic MOC-gel matrix facilitate fast photo-oxidation of tetraphenylethene and tune the emission to pure white light.

Remarkable diversity of the present anionic MOC to form hydrogels with range of molecular binders clearly suggest the opportunity of preparing highly efficient light-emitting hydrogels with desired optical output based on rationally designed chromophoric binders. Although the use of MOC white light gels as active layer components in real optoelectronic devices is not studied here, we can expect substantial progress of MOC gels in near future with detailed studies on efficiency and durability in real optoelectronics.

4. The authors might also comment further on links to metal organic framework gels, such as DOI:10.1038/nmat5050, and give an indication of how thermal and mechanical stability compares to other materials once dried.

Response:

We are thankful to the reviewer for this suggestion. The monoliths gel of HKUST-1 reported in DOI:10.1038/nmat5050 showed high thermal stability (ca. 330 °C) as comparable to the as-synthesized HKUST-1. HKUST-1 is a 3D porous framework extended by strong metal-carboxylate interactions and exhibit high thermal stability. However, supramolecular hydrogels are soft materials and often possesses low thermal and mechanical stability. Here, **MOC-G1** hydrogel is extended in 3D through weak non-covalent interaction (charge assisted H-bonding interaction) with ammonium cations and therefore is expected to show relatively low thermal stability compared to HKUST-1. This is also evident from TGA of **MOC-G1** xerogel which shows thermal stability up to ca. 185 °C and the thermal stability is almost

similar to the pristine **MOC** crystals. It is worth to be pointed that the **MOC** structural features and stability are retained even upon gelation.

We studied the mechanical stability of the **MOC-G1** hydrogel by rheology analysis (see the revised supporting information). Storage modulus (G') at high strain (%) was found to be 52.4 Pa (**Figure 1a**). Further, the stress versus strain plot (**Figure 1b**) suggested a value of 6.18 Pa and 17.8% as yield stress and yield strain values, respectively. However, studying the mechanical stability of the hydrogel after drying (via nano-indentation as done in DOI:10.1038/nmat5050) is beyond the scope of this work.

Figure 1. (a) Oscillatory strain measurements (frequency=1.0 rad/s) of **MOC-G1**, the squares (black) and circles (red) indicate storage (G') and loss modulus (G'') respectively, (b) The stress vs strain plot.

5. Abstract: Several small omissions. “Well defined metal-organic cubes (MOC) into”. “Like the”

Response:

We are thankful to the reviewer for pointing out these mistakes. We have corrected these small omissions throughout the manuscript.

6. I disagree with the authors assertion that “PXRD of 1 and xerogel showed no significant change, which suggests MOCs are intact”. Peaks have appeared/disappeared, shifted position and got stronger/reduced in intensity all over here.

Response:

We are thankful to the reviewer for highlighting this part. We agree with the reviewer that there are minor changes in the PXRD patterns **MOC-G1** xerogel compared to as-synthesized **MOC**. It is intended to say that the characteristic diffraction peaks of **MOC** are retained in the xerogel and structural features of **MOC** are restored after hydrogelation. To further support the stability of **MOC** after hydrogelation, we have carried out HRMS analysis on the **MOC-G1** xerogel (**Figure 2**). Peaks at $m/z = 2469.6475$ ($z = 1^-$), 1233.3665 ($z = 2^-$)

and 822.2319 ($z = 3^-$) corresponds to $[\{\text{Ga}_8(\text{ImDC})_{12}\}\{9\text{H}^+\}\{2\text{Na}^+\}\{\text{H}_2\text{O}\}]^-$, $[\{\text{Ga}_8(\text{ImDC})_{12}\}\{8\text{H}^+\}\{2\text{Na}^+\}\{\text{H}_2\text{O}\}]^{2-}$, and $[\{\text{Ga}_8(\text{ImDC})_{12}\}\{7\text{H}^+\}\{2\text{Na}^+\}\{\text{H}_2\text{O}\}]^{3-}$ moieties, respectively further confirms the stability of **MOC** after hydrogelation. Moreover, the shifting of peak position or change in intensity in PXRD is expected as in the as-synthesized **MOC** structure two cubes are connected by dimethyl ammonium cations, but in **MOC-G1** xerogel they are connected by ammonium binders also. We have rewritten the sentence “PXRD of **1** and xerogel showed no significant change, which suggests MOCs are intact” in the revised main manuscript.

Figure 2: HRMS of aqueous solution of **MOC-G1** xerogel showing peaks at $m/z = 2469.6475$ ($z = 1^-$), 1233.3665 ($z = 2^-$) and 822.2319 ($z = 3^-$).

7. The SI needs to be corrected. There are several figures that are separated from their titles, which is perhaps a PDF conversion problem. The capture for Fig. S2 should read cubes. The figure label for S10 should, perhaps, read minutes, not mints.

Response:

We are grateful to the reviewer for pointing out these mistakes. We have carefully gone through the supporting information and corrected/ modified accordingly.

Reviewer #2

The manuscript entitled “Binder driven self-assembly of metal-organic cubes towards functional hydrogels” demonstrates formation of hydrogels from metal-organic cube by using molecular binders. The authors cleverly showed interesting applications for the hydrogels produced via assembly, for example gel-chromatographic separation of cationic species from anionic counterparts, tuning emission color etc. The manuscript is of high quality in terms of

experimental procedure and applications. However the reviewer would like to mention a few points and seeking clarifications for others:

Response:

We are thankful to the reviewer for appreciating of our work.

1. The reviewer suggests including a detailed explanation of confirmation of intact MOCs in solution.

Response:

A detailed explanation of MOC stability is included in the revised manuscript and the same is given below. Solution stability of MOC was studied by HRMS; negative scan HRMS of MOC (aq.) showed three peaks (**Figure 3**) at $m/z= 2467.3295$, 1233.1622 and 822.5894 corresponding to anionic cubes associated with counter ions i.e. $[\{Ga_8(ImDC)_{12}\}\{9H^+\}\{2Na^+\}\{H_2O\}]^-$ ($z = 1^-$), $[\{Ga_8(ImDC)_{12}\}\{8H^+\}\{2Na^+\}\{H_2O\}]^{2-}$ ($z = 2^-$) and $[\{Ga_8(ImDC)_{12}\}\{7H^+\}\{2Na^+\}\{H_2O\}]^{3-}$ ($z = 3^-$), respectively. These clearly suggest the existence of anionic MOC in aqueous solution. It is noteworthy; from the aq. solution we got the single crystals of **1** for X-ray analysis. The free carboxylate oxygen atoms presence on the periphery of MOC ($z = 12^-$) makes it highly polar. Therefore water molecules readily solvate MOCs by H-bonding interaction with peripheral carboxylate oxygen.

Figure 3. Negative mode acquisition mode HRMS of the aqueous solution of **1** showing the peak at $m/z= 2467.3295$ ($z = 1^-$), 1233.1622 ($z = 2^-$) and 822.5894 ($z = 3^-$).

2. What is the effect of temperature on **MOC-G1** hydrogel?

Response:

The **MOC** hydrogels studied here are soft and gelation occurs at room temperature. Reversible gel-sol transition was observed after sonicating and resting the gel. However, on slow heating of the hydrogels (below 90 °C) release of water by evaporation was observed leading to formation of shrunken gels and then xerogel. So we believe that the hydrogel are not thermo-reversible.

3. The strength of **MOC-G1** hydrogel in terms of a storage modulus (G') should be given in the main text. Is it comparable with the existing hydrogels formed from metal-organic cages linked by molecular binders?

Response:

The rheological data explaining the storage modulus of **MOC-G1** is added to the main text and the manuscript is revised according to the reviewer's suggestion. Storage modulus (G') at high strain (%) was found to be 52.4 Pa for **MOC-G1** (**Figure 1a**). Further, the stress versus strain plot (**Figure 1b**) suggested a value of 6.18 Pa and 17.8% as yield stress and yield strain values, respectively. The previous reports [such as poly**MOC** gels in *Nat. Chem.* 8, 33–41 (2016)] are mainly polymeric gel. The C-C linkage connected polymer chains (PEG) with terminal coordinating groups are assembled with metal ions to form poly**MOC** gel which consists of in-situ formed metal-organic cages at the junction of cross-linked polymers. Hence, it is expected that they possess higher mechanical stability due to long polymer chains comprising the hydrogel. In contrast, **MOC-G1** is comprised of weak H-bonding interactions between preformed **MOCs** and small molecule binders; hence they are expected to have less mechanical stability compared to literature reports. However, we feel that the direct comparison of in-situ poly**MOC** gels and self-assembled **MOC** gels is not suitable.

4. Rheology results of **MOC-G5** should be included in supporting information.

Response:

As suggested by the reviewer we have included the rheology results of **MOC-G5** in the supporting information and discussed about it in the revised main manuscript. Oscillatory strain measurements (frequency=1.0 rad/s) of **MOC-G5** is shown in **Figure 4**.

Figure 4. Oscillatory strain measurements (frequency=1.0 rad/s) of **MOC-G5**.

5. Although application of the MOC-based gel for light harvesting application is novel, mixing different dyes such as rhodamine and tetraphenylethene in a gel matrix to get white light emission is already reported in literature. Several of fluorescent hydrogels are known to be used for photo patterning and white light emission. Therefore advantage of the present system over the existing methods/systems should be addressed.

Response:

The major challenge in preparing white-light-emitting material from mixture of chromophoric dyes is circumventing aggregation caused fluorescence quenching of chromophores in solid state or thin film state. Herein, this challenge was overcome by introducing MOC-gel matrix where anionic MOCs direct the self-assembly of the cationic chromophores while maintaining optimum inter-chromophoric distance and significantly reduce non-radiative energy loss. The main advantage of the MOC-gel matrix in preparing fluorescent hydrogels is its templating behavior to reduce aggregation caused quenching effects of dye molecules. Such approaches were previously demonstrated using amino clay supporting matrix by George *et al.* (*Adv. Mater.*, 2013, 25, 1713–1718. *Chem. Sci.*, 2015, 6, 6334-6340). The novelty of our light-harvesting system focused on using anionic MOC as a new type of supporting matrix for controlled assembly of chromophoric molecules. The defined geometry of MOC and the surface negative charge provide directional co-assembly of cationic tetraphenylethene, rhodamine and facilitate efficient energy transfer. Further, the structural constraints in tetraphenylethene induced by anionic MOC-gel matrix facilitate fast photo-oxidation of tetraphenylethene and tune the emission to pure white light. Moreover, due to high processability the hydrogel could be easily coated on flexible surface and photo-patterning or writing would be easier with these hydrogels compared to powder materials.

6. The concept is already reported in ref. 40-42

Response:

Same explanation as in question #1 of first reviewer

7. Use of the material for the column-chromatographic separation oppositely charged species is a novel concept and is well demonstrated.

Response:

We are thankful to the reviewer for appreciating our work.

8. In Figure (3) caption Spelling 'choloumn' should be corrected.

Response:

We are thankful to the reviewer to point out the mistake. We have corrected the spelling to "column" in the revised main manuscript.

9. In Figure 1, Nano-tube gel and nano-cube gels are transparent while others are opaque, justify.

Response:

The nanotube (**MOC-G1**), nanocube (**MOC-G4**) and nanosheet (**MOC-G3**) hydrogels are transparent while nano-bouquet (**MOC-G2**) and nanocube (**MOC-G5**) hydrogels are opaque. This is probably because **MOC-G1**, **MOC-G3** and **MOC-G4** took long time to form stable gel (8-10 hours for **MOC-G1** and 24 hours for **MOC-G3** and **MOC-G4**). However, **MOC-G2** is formed after 3-4 hours and **MOC-G5** hydrogel is formed immediately during sonication. Therefore, formation of **MOC-G1**, **MOC-G3** and **MOC-G4** is controlled by thermodynamics, whereas formation of **MOC-G2** and **MOC-G5** is kinetically controlled. It is known in literature that thermodynamically controlled gels are generally transparent and kinetically controlled gels are opaque in nature (*doi: 10.1016/j.msec.2014.06.017*).

10. Figure 2. Pretty images rectangular tubes were shown. I am wondering why they are rectangular in shape, whether the inner tube is hollow, what is the size of the inner channel (150-200 nm)? This value corresponds to the size of a cube? Please explain.

Response:

The nanotubular structures of **MOC-G1** are indeed hollow and the inner diameter is ca. 80-100 nm. The wall thickness was found to be 9-10 nm which is an assembly of 6-7 anionic **MOCs** (1.355 nm, crystallographically calculated) via H-bonding with NH_4^+ cations. We propose that the competitive interactions between anionic **MOC** and NH_4^+ (higher

concentration), Me_2NH_2^+ (low in concentration) in solution can result in anisotropic self-assembly to form nanotubular structures. We believe Me_2NH_2^+ is playing role to result in nanotubular structures of **MOC-G1** hydrogel.

11. Figure 3. Is there is any suggestive mechanism of tube growth (proposed scheme), especially from 1D tape to a hollow rectangular tube? Figure 3 b, and c are not convincing enough – it can be due to drying induced artifact too. Replace with better images, if available. The Chromatographic scheme is well done!

Response:

We propose that the competitive interactions between anionic **MOC** and NH_4^+ (higher concentration), Me_2NH_2^+ (low in concentration) in solution could result in anisotropic self-assembly to form nanotubular structures. We believe Me_2NH_2^+ is playing role to result in nanotubular structures of **MOC-G1** hydrogel. As per reviewer's suggestions we have replaced the Figure 3b and 3c with better FESEM images in main text and revised the manuscript accordingly. We want to mention that the **MOC-G1** hydrogel is formed after 8 hours and we have collected the FESEM images of the (**MOC**+ NH_4^+) solution after every 2 hours. Each time before recording FESEM we have dispersed the sample in 500 μl ethanol and drop-casted on silica substrate. As the FESEM images at 2h, 4h and 6h are obtained before complete hydrogel formation, therefore we believe that the 1D tape-like structures and partially-grown nanotubes are not resulting from drying induced artifact.

We are thankful to the reviewer for appreciating chromatographic scheme.

12. Figure 4. What drives the mechanism of growth into different morphologies with different binders? Among the three binders, is there any specificity towards one binder, if a mixture was used?

Response:

We are thankful to the reviewer for the critical observations and suggestions. We believe that the difference in structural features (geometry, size, shape and flexibility) of molecular binders affect the morphology of the nanostructure formed after self-assembly with **MOC**. To understand the specificity of anionic **MOC** towards a binder, we have carried out gelation of **MOC** with mixture of binders. 0.01 M aqueous solutions of ammonium ions (NH_4^+), N-(2-aminoethyl)-1,3-propanediamine (AEPD), guanidine hydrochloride (gua.HCl), and β -alanine (β -ala) were prepared. The following combinations of binder solutions were added into **MOC** solution to prepare mixed binder hydrogels.

- i) 100 μl NH_4^+ + 100 μl AEPD + 100 μl gua.HCl (For **MOC-GM1** hydrogel)
- ii) 100 μl AEPD + 100 μl gua.HCl + 100 μl β -ala (For **MOC-GM2** hydrogel)
- iii) 100 μl NH_4^+ + 100 μl gua.HCl + 100 μl β -ala (For **MOC-GM3** hydrogel)
- iv) 100 μl NH_4^+ + 100 μl AEPD + 100 μl β -ala (For **MOC-GM4** hydrogel)

Figure 5. Picture of **MOC-GM1**, **MOC-GM2**, **MOC-GM3** and **MOC-GM4**, respectively (from left to right).

Figure 6. a-f) FESEM images of **MOC-GM1** xerogel showing co-existence of nanotubes, needle-like and sheet-like nanostructures, b) FESEM image of **MOC-GM1**. The red circles highlight the bunches of nanotubes and needle-like nanostructures, c-e) FESEM images showing the presence of nanotubes and needle-like nanostructures, f) FESEM image showing co-existence of sheet-like and needle-like nanostructures. The blue circle highlights the presence of nano-sheet.

Interestingly, in all cases we have observed the formation of stable opaque hydrogels (in 4 h) and are named as **MOC-GM1**, **MOC-GM2**, **MOC-GM3** and **MOC-GM4**, respectively (**Figure 5**). FESEM images of **MOC-GM1** xerogel showed formation of nanotubes, needles and sheet-like nanostructures (**Figure 6**). The co-existence of three different morphologies indicates that NH_4^+ , AEPD and gua.HCl individually drive the self-assembly of **MOCs** to respective nanostructures. However, the number of nanotubes and nanoneedles are higher

compared to the nanosheet. This may indicate higher affinity of NH_4^+ /AEPD to interact with anionic MOCs than gua.HCl. Similarly, FESEM images of MOC-GM2 xerogel, containing AEPD, gua.HCl and β -ala as binders show the co-existence of needle-like, sheet-like

Figure 7. a-f) FESEM images of MOC-GM2 xerogel showing co-existence of nanocubes, needle-like and sheet-like nanostructures, a-b) FESEM image of MOC-GM2 showing the nanocubes, c) FESEM image showing the presence of needle-like nanostructures, d) FESEM image showing co-existence of needle-like nanostructures (blue circle), nanocubes (red circle) and sheet-like nanostructures (yellow circle), e-f) Showing the presence of needle-like nanostructures and nanocubes.

Figure 8. a-f) FESEM images of MOC-GM3 xerogel showing co-existence of nanotube, sheet-like nanostructures and nanocube, a) The low resolution FESEM image showing the bunches of nanotubes highlighted by red circle, b-d) FESEM images showing co-existence of nanotubes and nanosheets. Sheet-like nanostructures are highlighted by yellow circles, e-f) The FESEM images showing co-existence of nanotubes and nanocube.

Figure 9. a-f) FESEM images of **MOC-GM4** xerogel showing the co-existence of nanotube and needle-like nanostructures.

nanostructures and nanocubes, with equal distribution (**Figure 7**). FESEM images of **MOC-GM3** xerogel, containing NH_4^+ , gua.HCl and β -ala as binders show the presence of nanotubes, nanocubes and sheet-like nanostructures with equal distribution (**Figure 8**). Surprisingly, FESEM images of **MOC-GM4** xerogel, containing NH_4^+ , AEPD and β -ala as binders show the co-existence of nanotubes, needle-like nanostructures and few nanocubes (**Figure 9**). This is probably because NH_4^+ and AEPD have more affinity towards **MOCs**. In most cases, three different morphologies are present in one system and show nearly equal distribution.

Figure 10. Isothermal titration calorimetry analysis of **MOC** with NH_4^+ (left) and **AEPD** (right) in water at 25 °C. The upper panel shows the raw data curve, lower panel shows the fitted integrated ITC data curve.

Figure 11. Isothermal titration calorimetry analysis of **MOC** with β -ala (left) and gua.HCl (right) in water at 25 °C. The upper panel shows the raw data curve, and the lower panel shows the fitted integrated ITC data curve.

To further know the binding affinity of molecular binders to **MOC**, we performed isothermal titration calorimetric (ITC) experiments. The representative calorimetric titrations of NH_4^+ , AEPD, β -ala and gua.HCl into the solution of **MOC** at 25 °C are shown in **Figure 10** and **11**. With incremental addition of NH_4^+ , AEPD, β -ala and gua.HCl aliquots to aqueous solution of **MOC**, exothermic heat changes are observed. The binding affinity values (K_a), stoichiometry (N), enthalpy (ΔH), entropy (ΔS) and free energy (ΔG) changes are listed below in Table 1. For all molecular binders the K_a values are obtained in the order of 10^4 M^{-1} and varies as $\text{AEPD} > \text{NH}_4^+ > \beta\text{-ala} > \text{gua.HCl}$. This indicates AEPD, NH_4^+ and β -ala have comparable binding affinity which is three times higher than gua.HCl. The result is also in good agreement with the morphological distributions obtained from **MOC-GM1**, **MOC-GM2**, **MOC-GM3** and **MOC-GM4**. Moreover, AEPD and NH_4^+ have higher binding affinity than β -ala which also explains why we observe more nanotubes and nano-needles in the FESEM images of **MOC-GM4**.

Table 1: The binding affinity values (K_a), stoichiometry (N), enthalpy (ΔH), entropy (ΔS) and free energy (ΔG) of different binders.

Binder	$K_a (\text{M}^{-1})$	N	$\Delta H (\text{kcal mole}^{-1})$	$T\Delta S (\text{kcal mole}^{-1})$	$\Delta G (\text{kcal mole}^{-1})$
NH_4^+	6.65×10^4	10.5	-19.61	-1.0925	-18.5175
AEPD	6.76×10^4	9.86	-18.90	-1.0325	-17.8675

β -ala	6.50×10^4	9.1	-22.86	-1.3650	-21.4950
Gua.HCl	2.73×10^4	7.96	-26.53	-1.7175	-24.8125

13. In summary, this is an interesting piece of work to create hydrogels from discrete nanocubes via self-assembly. The authors elegantly showed multiple possible applications of such gels. However, several clarifications are needed and the reviewer suggest for a major revision of the manuscript.

Response:

We are grateful to the reviewer for maintaining our work as “interesting piece of work” and suggested the major revision of the manuscript. The suggestions and comments of the reviewer have really helped to improve the overall quality of the manuscript.

Reviewer #3

In this manuscript, the authors have demonstrated the charge-assisted hydrogen bond driven self-assembly formation of hydrogels from a Ga^{3+} anionic metal-organic cube (MOC) and cationic binders in water. The manuscript is well written and to the point and has some very interesting results. The focus herein is on the material site and the authors observed that the morphology of resulting hydrogels is depends on the size, shape and geometry of the molecular binder used to make the gels. The nanotubular hydrogel obtained from the combination of MOC and NH_4^+ was used as stationary phase for the selective separation of cationic dyes (nile blue and acridine orange) from their anionic counterpart (sulforhodamine G). Furthermore, authors have also made few luminescent hydrogels and employed them as light harvesting antenna to tune the emission colour to develop white light emitting devices.

The use of charge-assisted hydrogen bonding (CAHB) interaction to generate self-assembled structures and soft-materials is not something new (as the authors point out), and there are several reports of using MOC as building blocks to generate self-assembled structures and materials. Though the idea of using CAHB interaction as primary driving force to generate hydrogels is not novel, the morphological analysis and the application of hydrogels for the chromatographic separations of dyes is interesting and this is the strength of this manuscript. Overall the work described in this article will be of interest to a wide range of researchers and thus I am recommending that it can be considered for publication subject to a revision.

Response:

We are grateful to the reviewer for appreciating our work and for overall comments about the manuscript.

1. It is known that even free carboxylate ions form self-assembled nanomaterials with interesting morphological features upon binding with cationic molecular binder through charge-assisted hydrogen bonding with interactions. Therefore, I request the authors to analyse the morphological features of 4,5-imidazoledicarboxylic acid, that was used to make MOC, in presence of molecular binders. This will help the reader to understand the role of anionic metal-organic cages with confined cavities to form the self-assembled supramolecular hydrogels.

Response:

We are thankful to the reviewer for suggesting this important experiment. As suggested, we have carried out experiments to analyze the self-assembly of 4,5-imidazoledicarboxylic acid (H_3ImDC) in presence of molecular binder (NH_4^+). In three different glass vials 10, 15 and 20 mg of H_3ImDC are taken and 1mL water is added in each. 18 μ l NEt_3 is added to each vial to deprotonate and dissolve H_3ImDC . Next, aqueous ammonia solution (100 μ L) is added to each vial and the resulted solutions are kept undisturbed at room temperature for 4 days. As shown in **Figure 12a**, no hydrogel formation was observed. Further, TEM images of the solution did not show any particular morphology; rather random flaky particles were observed (**Figure 12b-d**). These observations clearly suggest the importance of anionic **MOC** in driving the self-assembly with different molecular binders to form hydrogels of different morphology. We are thankful to the reviewer for this critical question, inclusion of this information in the manuscript indeed help the readers to understand the role of **MOC** for the formation of self-assembled supramolecular hydrogels.

Figure 12. (a) Photograph of the aq. solution of 4,5-imidazoledicarboxylic acid containing NH_4^+ cations as molecular binders, (b)-(d) TEM images of the same solution.

2. The stability of MOC is a series concern of the work described in this article. How stable the MOC is after the hydrogel formation and dye encapsulation and photoirradiation?

Response:

The anionic MOCs are indeed stable in aqueous solution and it is evident from the HRMS analysis as explained in the response to question no 1 of reviewer#2. To know the stability of MOC after gel formation, dye encapsulation and photo-irradiation we have recorded HRMS after each step. HRMS of MOC-G1 xerogel shows the highest intensity peak at $m/z=2469.6475$ ($z=1^-$), 1233.3665 ($z=2^-$) and 822.2319 ($z=3^-$) corresponds to

Figure 13: HRMS of aqueous solution of MOC-G1 xerogel showing peaks at $m/z=2469.6475$ ($z=1^-$), 1233.3665 ($z=2^-$) and 822.2319 ($z=3^-$).

Figure 14: HRMS of NB encapsulated MOC-G1 xerogel showing peaks at $m/z=821.5833$ and 1233.3417

$[\{Ga_8(ImDC)_{12}\}\{9H^+\}\{2Na^+\}\{H_2O\}]^-$, $[\{Ga_8(ImDC)_{12}\}\{8H^+\}\{2Na^+\}\{H_2O\}]^{2-}$, and $[\{Ga_8(ImDC)_{12}\}\{7H^+\}\{2Na^+\}\{H_2O\}]^{3-}$ moieties, respectively (**Figure 13**). This confirms the stability of MOC in **MOC-G1** hydrogel. HRMS of Nile blue (NB) encapsulated **MOC-G1** shows peaks at $m/z = 821.5833$ ($z = 3^-$) and 1233.3417 ($z = 2^-$) corresponding to $[(2Na^+)(7H^+)\{Ga_8(ImDC)_{12}\}(H_2O)]^{3-}$ and $[(2Na^+)(8H^+)\{Ga_8(ImDC)_{12}\}(H_2O)]^{2-}$ moieties respectively, thus confirming the stability of **MOCs** in NB encapsulated **MOC-G1** hydrogel (**Figure 14**). The negative scan HRMS of the photo-irradiated **MOC** shows peaks at $m/z = 1233.2622$ ($z = 2^-$), corresponding to $[(2Na^+)(8H^+)\{Ga_8(ImDC)_{12}\}(H_2O)]^{2-}$ moiety respectively, indicating the stability of MOC after photo-irradiation (**Figure 15**). Moreover, the positive scan HRMS after photo-irradiation shows the peaks corresponding to TPEQA, Rh6G and DPPQA, indicating their presence in the gel (shown in revised supporting information). Moreover, PXRD analysis of **MOC-G1** and **MOC-G5** xerogels show the presence of Bragg's reflections similar to **MOC**, further confirming the stability of MOC in **MOC-G1** and **MOC-G5** and after photo-irradiation.

Figure 15: HRMS of photo-irradiated **Rh6G_{0.08%}@MOC-G5** showing peaks at $m/z = 1233.2622$

3. It is interesting to see that MOC forms hydrogel of different morphology depending on the size, shape and geometry of molecular binder. Is MOC form hydrogel in presence of more than one binder? Any change in the morphological features in presence of more than one binder at once?

Response:

We are thankful to the reviewer for the interesting suggestions. The similar questions were also raised by the reviewer#2. We have tried to prepare MOC-hydrogels by using all possible combinations of three molecular binders out of four, NH_4^+ , N-(2-aminoethyl)-1,3-propanediamine (AEPD), guanidine hydrochloride (gua.HCl), and β -alanine (β -ala). The followings are the four possible combinations:

i) NH_4^+ + AEPD + gua.HCl

ii) AEPD + gua.HCl + β -ala

iii) NH_4^+ + gua.HCl + β -ala

iv) NH_4^+ + AEPD + β -ala

In all the cases we observed formation of stable hydrogels which are named as **MOC-GM1**, **MOC-GM2**, **MOC-GM3** and **MOC-GM4**. We have characterized the morphologies of the hydrogels by FESEM analysis. Please see the details experimental results and responses of the question number 12 of reviewer#2

4. MOC-G1 was used for gel chromatographic separation of cationic dye molecules from the anionic counterpart. MOC-G1 was regenerated after washing the HCl. The size and geometry of cationic dye is larger than the NH_4^+ . Is MOC-G1 is exhibit nano-tube like structure after the adsorption and separation of dyes? How stable the MOC-G1 after washing with HCl?

Response:

We are thankful to the reviewer for the critical observations for our work and corresponding suggestions. We have carried out morphology studies of **MOC-G1** after adsorption of the Nile blue (NB), as shown in **Figure 16**, TEM images of **NB@MOC-G1** shows the presence of nanotubular structures suggesting that the morphology of **MOC-G1** is unchanged even after dye adsorption. We proposed that the cationic dye molecules get

Figure 16. TEM images of the **MOC-G1** after adsorption of Nile blue.

Figure 17. TEM images of the **MOC-G1** after separation of Nile blue by washing with MeOH.

Figure 18. FESEM images of the precipitate formed after addition of aq.HCl to the dye adsorbed MOC-G1.

Figure 19. The PXRD pattern of the precipitate formed after degrading the MOC-G1 hydrogel with 0.1N aq. HCl.

attached on the surface of nanotube by electrostatic interaction. Therefore, no significant changes are expected after dye adsorption. After dye separation the hydrogel column was repeatedly washed with MeOH to remove absorbed Nile blue (**NB**). We recorded the TEM image of the hydrogel column after removal of **NB**, it reveals the retention of nanotube morphology (**Figure 17**). These observations confirm the stability of nanotube after dye adsorption and separation. After chromatographic separation the hydrogel-column is washed with 0.1 N aq. HCl which results in an immediate precipitation of **MOCs**. This occurs

mainly due the protonation of the surface carboxylate groups of **MOCs** by HCl and disruption of the H-bonding interactions between **MOC** and NH_4^+ . Interestingly, the FESEM images of the precipitate obtained show nanocube morphology (**Figure 18**) and the formation of hydrogel was observed when aq. NH_3 is added to the precipitate. Further, PXRD of the precipitate shows all the characteristic peaks of the as-synthesized **MOC** (**Figure 19**). This indicates the anionic **MOCs** are stable after dye adsorption, after pH-responsive precipitation and therefore can be recycled to hydrogels for further use in chromatographic separation.

5. The morphology and stability of **MOC-G5** gel also need to be checked after the photoirradiation and rhodamine encapsulation.

Response:

The morphology of the **MOC-G5** remains same (nanocube) after photoirradiation and rhodamine encapsulation (**Figure 20**). The HRMS analysis of photo-irradiated **MOC-G5** indicates the stability of **MOC** under photo-irradiation. The negative scan HRMS of the photo-irradiated **Rh6G_{0.08%}@MOC-G5** shows peaks at $m/z= 1233.2622$ ($z = 2^-$), corresponding to $[(2\text{Na}^+)(8\text{H}^+)\{\text{Ga}_8(\text{ImDC})_{12}\}(\text{H}_2\text{O})]^{2-}$ moiety respectively, indicating the

Figure 20. TEM images of **Rh6G_{0.08%}@MOC-G5** hydrogel after 5 min of photo-irradiation.

stability of **MOC** after photo-irradiation (**Figure 15**). Moreover, the positive scan HRMS after photo-irradiation shows the peaks corresponding to TPEQA, Rh6G and DPPQA, indicating their presence in the gel.

Finally, we are thankful to all the reviewers for their critical observations, positive comments and suggestions which have helped to improve the quality of the manuscript significantly.

REVIEWERS' COMMENTS:

Reviewer #1 (Remarks to the Author):

The authors have addressed the majority of my points and comments.

Reviewer #2 (Remarks to the Author):

The queries and concerns raised by reviewers were well addressed and corresponding corrections has been made appropriately by the authors. In my opinion the manuscript "Binder driven self-assembly of metal-organic cubes towards functional hydrogels" is now acceptable to Nature Communications.

Reviewer #3 (Remarks to the Author):

This revised version of the article on "Binder driven self-assembly of metal-organic cubes towards functional hydrogel" is much improved from its original submission. Authors have worked heavily to improve the technical qualities and satisfactorily addressed all the comments and concerns raised by the reviewers. Adequate changes and additions are incorporated in the revised manuscript. I am very much satisfied with the revision and this version is lives up to the standard for publication and will be of interest to a wide range of researchers. Therefore, in my opinion, this revised manuscript could be considered for publication in its current form.

Point-by-point reply to referees for the manuscript 'NCOMMS-17-33348-A'

Reviewer#1

1. The authors have addressed the majority of my points and comments.

Response: We are thankful to the reviewer for accepting the revised manuscript.

Reviewer#2

1. The queries and concerns raised by reviewers were well addressed and corresponding corrections has been made appropriately by the authors. In my opinion the manuscript "Binder driven self-assembly of metal-organic cubes towards functional hydrogels" is now acceptable to Nature Communications.

Response: We are thankful to the reviewer for accepting the revised manuscript and recommending for publication in Nature Communications.

Reviewer#3

1. This revised version of the article on "Binder driven self-assembly of metalorganic cubes towards functional hydrogel" is much improved from its original submission. Authors have worked heavily to improve the technical qualities and satisfactorily addressed all the comments and concerns raised by the reviewers. Adequate changes and additions are incorporated in the revised manuscript. I am very much satisfied with the revision and this version is lives up to the standard for publication and will be of interest to a wide range of researchers. Therefore, in my opinion, this revised manuscript could be considered for publication in its current form.

Response: We are thankful to the reviewer for appreciations and encouraging comments and also recommending for publication in its current form.

Finally, we are thankful to all the reviewers for their comments and suggestions which have immensely helped to improve the overall quality of the manuscript.